# Large Language-Geometry Model: When LLM meets Equivariance

**Zongzhao Li** [1 2 3]   **Jiacheng Cen** [1 2 3]   **Bing Su** [1 2 3]   **Tingyang Xu** [4 5]   **Yu Rong** [4 5]   **Deli Zhao** [4 5]   **Wenbing Huang** [1 2 3]

## Abstract

Accurately predicting 3D structures and dynamics of physical systems is crucial in scientific applications. Existing approaches that rely on geometric Graph Neural Networks (GNNs) effectively enforce E(3)-equivariance, but they often fail in leveraging extensive broader information. While direct application of Large Language Models (LLMs) can incorporate external knowledge, they lack the capability for spatial reasoning with guaranteed equivariance. In this paper, we propose EquiLLM, a novel framework for representing 3D physical systems that seamlessly integrates E(3)-equivariance with LLM capabilities. Specifically, EquiLLM comprises four key components: geometry-aware prompting, an equivariant encoder, an LLM, and an equivariant adapter. Essentially, the LLM guided by the instructive prompt serves as a sophisticated invariant feature processor, while 3D directional information is exclusively handled by the equivariant encoder and adapter modules. Experimental results demonstrate that EquiLLM delivers significant improvements over previous methods across molecular dynamics simulation, human motion simulation, and antibody design, highlighting its promising generalizability.

## 1 Introduction

Accurately predicting 3D structures/dynamics of physical systems remains a fundamental challenge in physics and biology. Typical tasks such as molecular dynamics simulation (Hollingsworth & Dror, 2018) and antibody de-

sign (Tiller & Tessier, 2015) require not only a deep understanding of complex spatial geometry but also the preservation of E(3)-equivariance — ensuring predictions transform correspondingly with input rotations, reflections and translations (Batzner et al., 2022; Huang et al., 2022). From the machine learning perspective, E(3)-equivariant models are more powerful than their non-equivariant counterparts, as they are inherently generalizable across arbitrary coordinate systems when modeling physical systems. To achieve equivariance, current approaches primarily rely on geometric Graph Neural Networks (GNNs) (Wu et al., 2024; Kong et al., 2022; Li et al., 2025a). Despite their fruitful progress, these models often lack the ability to leverage external domain knowledge and broader contextual information, such as task-specific instructions and expert-curated guidance, hindering further performance enhancement.

Recently, Large Language Models (LLMs) have demonstrated remarkable success across a wide range of applications, owing to their large-scale pretraining on extensive datasets and their substantial model size (Sun et al., 2025a; Li et al., 2025c; Yang et al., 2025a; Sun et al., 2025b; Bian et al., 2025; Liu et al., 2025b; Li et al., 2025b; Xu et al., 2025; Sun et al., 2025c; Liang et al., 2025; Yang et al., 2025b). It is well known that LLMs can not only understand and generate text but also excel at integrating and leveraging scientific knowledge (Liu et al., 2025a; Jablonka et al., 2024; Wang et al., 2023b). For instance, LLMs can comprehend fundamental chemical concepts and molecular structural characteristics (Guo et al., 2023). More significantly, based on our results, we speculate that LLMs' flexibility in prompt engineering enables the development of tailored instructions that better leverage their capabilities, producing outputs more precisely suited to the task.

A natural idea is to directly employ LLMs for modeling 3D physical systems. However, this approach *fails to* yield satisfactory results in practice. A key limitation is that LLMs are trained to process ordered and discrete text tokens, restricting their ability to directly comprehend unordered and continuous data in 3D space. One possible solution is to adapt existing multimodal LLM architectures, such as LLaVA (Liu et al., 2024b), by treating 3D structures as a separate modality and simply replacing the image encoder with a geometric GNN. However, this naive adaptation fails to satisfy the E(3)-equivariance requirement. Since geometric

---

[1]Gaoling School of Artificial Intelligence, Renmin University of China [2]Beijing Key Laboratory of Research on Large Models and Intelligent Governance [3]Engineering Research Center of Next-Generation Intelligent Search and Recommendation, MOE [4]DAMO Academy, Alibaba Group, Hangzhou, China [5]Hupan Lab, Hangzhou, China. Correspondence to: Wenbing Huang <hwenbing@126.com>, Yu Rong <yu.rong@hotmail.com>.

*Proceedings of the 42$^{nd}$ International Conference on Machine Learning*, Vancouver, Canada. PMLR 267, 2025. Copyright 2025 by the author(s).

GNNs produce both invariant features and equivariant coordinates, passing these outputs through an LLM inevitably compromises equivariance. Although the canonicalization approach (Puny et al., 2022; Mondal et al., 2023) enables the application of non-equivariant LLM to equivariant tasks, our experiments empirically show that the resulting performance is still suboptimal. *Therefore, it is non-trivial to integrate the strengths of both LLMs and geometric GNNs while maintaining essential geometric properties and ensuring strong performance.*

To this end, this paper introduces EquiLLM, a novel framework for representing 3D physical systems that seamlessly integrates E(3)-equivariance with LLM capabilities. Equi-LLM is carefully designed and comprises four core modules (see Figure 1): geometry-aware prompting, an equivariant encoder, an LLM, and an equivariant adapter. A key insight of EquiLLM in maintaining equivariance lies in its innovative design: the LLM guided by the instructive prompt serves as a sophisticated invariant feature processor, while 3D directional information is exclusively handled by the equivariant encoder and adapter modules. Specifically, to fully activate the spatial reasoning capabilities of LLMs, we introduce the geometry-aware prompts containing invariant geometric information, including task description, input feature description and statistical information. Then, EquiLLM employs a geometric GNN as a domain-specific encoder to effectively model and extract 3D representations of input systems. Subsequently, to satisfy the equivariance constraint, the input to the LLM are strictly limited to the prompt and invariant features derived from the equivariant encoder's output. The LLM-generated outputs are subsequently combined with the 3D equivariant vectors produced by the equivariant encoder and fed into an equivariant adapter for information fusion. EquiLLM ultimately produces both invariant labels and equivariant coordinates required by downstream applications.

To sum up, our main contributions are threefold:

- To the best of our knowledge, we present the first investigation into modeling 3D physical systems by integrating LLMs with geometric GNNs, aiming to combine the strengths of both approaches.

- We present EquiLLM, a novel framework that is meticulously designed to permit E(3)-equivariance and instill 3D spatial reasoning into LLMs' powerful capabilities.

- We conduct extensive experiments on diverse tasks of molecular dynamics simulation, human motion simulation and antibody design. The results show that our method achieves superior performance, attaining state-of-the-art results in nearly all metrics.

## 2  Related Work

**Geometric GNNs.**  Geometric GNNs have achieved significant success across a wide range of scientific applications by leveraging physical symmetries in 3D space (Han et al., 2024; Xu et al., 2024; Wang et al., 2025a; Li et al., 2024d;e;c; Yan et al., 2025; Wu et al., 2025; Lin et al., 2025; Han et al., 2025; Liu et al., 2025c; Wang et al., 2025b). Among them, tensor-product-based models (Thomas et al., 2018; Fuchs et al., 2020; Brandstetter et al., 2021; Batatia et al., 2022; An et al., 2025; Xie et al., 2025) excel at capturing interactions between steerable features of different degrees but are computationally expensive. In contrast, scalarization-based models (Satorras et al., 2021; Schütt et al., 2021; Huang et al., 2022; Aykent & Xia, 2023; Zhang et al., 2024; 2025) focus on constructing invariant scalars (*e.g.* norms and inner products), which offer both efficiency and expressiveness. This approach has been further extended by spherical-scalarization methods (Frank et al., 2024; Cen et al., 2024; Aykent & Xia, 2025). Well-designed scalar features have also proven effective in improving performance in applications (Wang et al., 2023a; Zhou et al., 2023; Wang et al., 2024; Battiloro et al., 2024; Yue et al., 2024; 2025). For dynamics simulations, ESTAG (Wu et al., 2024) enhances trajectory prediction using frequency cross-correlations, while SEGNO (Liu et al., 2024c) reduces roll-out errors by integrating scalars into neural operator learning. In biological modeling, MEAN (Kong et al., 2022) improves antibody representation via a multi-channel scalar attention mechanism, and GeoAB (Lin et al., 2024) generalizes this to capture higher-order atomic interactions. Motivated by these successes, we investigate whether incorporating LLMs into scalar design can further enhance geometric models.

**LLM + GNN.**  LLMs with rich knowledge are being widely transferred and applied across multiple domains to enhance model capabilities (Singhal et al., 2023; Song et al., 2025; Wang et al., 2025c; Singhal et al., 2025). Numerous excellent works have emerged in combining GNNs with LLMs for scientific applications. ChemLLMBench (Guo et al., 2023) tests LLM's understanding, reasoning, and explaining capabilities on various chemical tasks using in-context learning. Prot2Text (Abdine et al., 2024) integrates protein sequence, structure, and textual annotations into an encoder-decoder framework composed of GNN and LLM to predict protein functions. MoleculeSTM (Liu et al., 2023a) uses a contrastive learning paradigm to align molecular graphs and textual descriptions in the semantic space, thereby learning better feature representations. MolCA (Liu et al., 2023b) employs Q-Former (Li et al., 2023) as a cross-modal projector to align the feature spaces of graph encoder and language encoder, enhancing performance in molecule captioning tasks. The aforementioned methods enhance interactions between GNNs and LLMs through various paradigms and yield promising results. However, exploring such LLM-

GNN integration paradigms for 3D structural data tasks, such as 3D structure generation and dynamic trajectory simulation, remains a relatively unexplored frontier. The EquiLLM framework we propose in this paper integrates LLMs with Geometric GNNs that embed spatial symmetry constraints and has been effectively validated across datasets from both physical and biological domains.

**3D Structure Tokenization.** As a critical step in methods employing language models for structural tasks, the tokenization of 3D structures has become an actively advancing research frontier, with numerous studies demonstrating its successful implementation. ESM3 (Hayes et al., 2025) encodes 3D atomic structures into discrete tokens, enabling seamless integration with sequences and functions that are similarly tokenized. This unified representation allows for coherent structural reasoning within a shared latent space. ProSST (Li et al., 2024a) tokenizes 3D structures by first serializing them into local structures at the residue level before encoding them into dense vector space. BindGPT (Zholus et al., 2025) employs an XYZ representation for structures, where the 3D coordinates of each atom are expressed as textual entries per line. Geo2Seq (Li et al., 2024b) converts 3D structures into 1D discrete sequences through canonical labeling and invariant spherical representations. CHEAP (Lu et al., 2024) utilizes FSQ (Mentzer et al., 2024) to encode 3D structures into a quantized latent space. However, such tokenization approaches may inherently carry risks of information loss. To address this, EquiLLM employs a dual-pathway architecture where the Geometric GNN processes 3D structure while the LLM handles discrete textual tokens, thereby preserving information integrity.

# 3 Method

In this section, we first introduce the preliminaries related to geometric modeling in § 3.1. Next, in § 3.2, we present the proposed framework EquiLLM. Finally, in § 3.3 and 3.4, we describe how EquiLLM is applied to two representative tasks (*e.g.* dynamic simulation and antibody design). The overview of our EquiLLM is illustrated in Fig. 1.

## 3.1 Preliminaries, Notations and Definitions

Physical systems (such as molecules) can be naturally modeled with geometric graphs. We represent each static system as a geometric graph $\mathcal{G} = (\mathcal{V}, \mathcal{E})$, where each node $v_i$ in $\mathcal{V}$ is associated with an invariant feature $\boldsymbol{h}_i \in \mathbb{R}^c$ (*e.g.* atom type) and an 3D equivariant vector $\vec{\boldsymbol{x}}_i \in \mathbb{R}^3$ (*e.g.* atom coordinates); each edge (*e.g.* chemical bonds) denotes the connectivity between nodes. Apart from modeling static systems, we explore dynamic systems, focusing on constructing geometric graphs across different time steps. The details will be thoroughly discussed in § 3.3 and 3.4. In the following sections, we use the matrices $\vec{\boldsymbol{X}} \in \mathbb{R}^{N \times 3}$

and $\boldsymbol{H} \in \mathbb{R}^{N \times c}$ to denote the sets of node coordinates and invariant features of the geometric graph $\mathcal{G}$.

**Task Formulation.** Here, we provide a general form of our task and will elaborate specific applications including dynamic simulation and antibody design in § 3.3 and 3.4. Given the input geometric graph $\mathcal{G}^{\text{in}}$, our goal is to find a function $\phi$ to predict the output $\mathcal{G}^{\text{out}}$. This process can be formally delineated as:

$$\mathcal{G}^{\text{out}} = \phi(\mathcal{G}^{\text{in}}). \tag{1}$$

Meanwhile, since we introduce LLMs into our framework, we will further construct task-specific prompts to guide the extraction of relevant domain knowledge from LLMs, recasting our task as:

$$\mathcal{G}^{\text{out}} = \phi(\mathcal{G}^{\text{in}}, \boldsymbol{P}), \tag{2}$$

where $\boldsymbol{P}$ denotes the prompt.

**Equivariance.** It is crucial to emphasize that in the tasks above, the function $\phi$ must satisfy $\text{E}(3)$ symmetries of physical laws (Han et al., 2024). Specifically, if arbitrary translations, reflections, or rotations are applied to the input coordinate matrix $\vec{\boldsymbol{X}}^{\text{in}}$, the output coordinate matrix $\vec{\boldsymbol{X}}^{\text{out}}$ should undergo the corresponding transformation.

## 3.2 Large Language-Geometry Model

In this section, we provide a meticulous description of our model EquiLLM, which consists of three main components: Equivariant Encoder, LLM, and Equivariant Adapter. Unlike existing works (Gruver et al., 2024) that applys an LLM to predict the 3D coordinates directly, EquiLLM leverages an LLM to acquire broader scientific domain knowledge while employing geometric GNNs for precise modeling of 3D structures. These two components are seamlessly integrated through an equivariant adapter, achieving superior predictive performance without compromising $\text{E}(3)$-equivariance.

**Equivariant Encoder.** The Equivariant Encoder is a domain-specific equivariant model, which can be any suitable equivariant model from the relevant field. The model takes the graph $\mathcal{G}^{\text{in}} = (\mathcal{V}^{\text{in}}, \mathcal{E}^{\text{in}})$ as input, performing initial encoding and embedding of geometric information, and outputs a processed geometric graph $\mathcal{G}' = (\mathcal{V}', \mathcal{E}')$, This process can be formally defined as:

$$\mathcal{G}' = \phi_e(\mathcal{G}^{\text{in}}), \tag{3}$$

where $\phi_e$ can be any equivariant model, used to jointly model the geometric relationships between $\vec{\boldsymbol{X}}^{\text{in}}$ and $\boldsymbol{H}^{\text{in}}$ features across different nodes, resulting in processed features $\vec{\boldsymbol{X}}'$ and $\boldsymbol{H}'$.

Since LLMs are not naturally equivariant, directly feeding $\vec{\boldsymbol{X}}'$ into an LLM would likely undermine the intrinsic

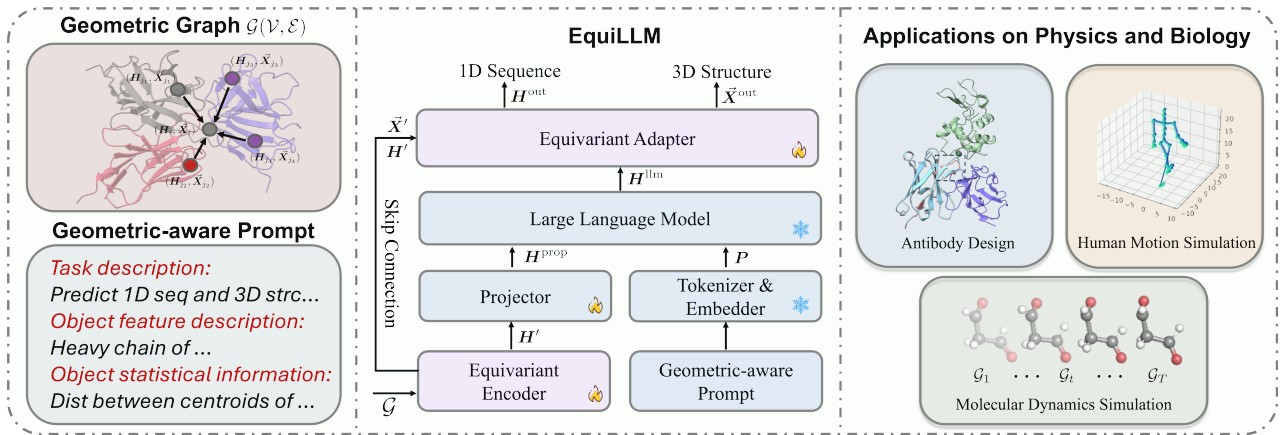

Figure 1: The overall framework of EquiLLM. Given a geometric graph $\mathcal{G} = (\mathcal{V}, \mathcal{E})$ as input, EquiLLM initially employs an Equivariant Encoder to derive processed features $\vec{X}'$ and $H'$. The features $H'$ are first projected through a projector, then concatenated with prompt features $P$ in a task-specific manner. This concatenated vector is subsequently fed into an LLM. The output features $H^{\text{llm}}$ from the LLM, alongside the previously obtained processed features $\vec{X}'$ and $H'$, are then passed into an Equivariant Adapter. The Equivariant Adapter then generates the final outputs, including the vector $\vec{X}^{\text{out}}$ for equivariant tasks and the feature $H^{\text{out}}$ for invariant tasks. The blue `module` means the invariant module, while the purple `module` means the equivariant module.

equivariance of the overall architecture. Thus, in contrast to existing works, we convey the invariant features $H'$ to the LLM, but pass the equivariant matrix $\vec{X}'$ to the subsequent Equivariant Adapter via a skip connection. Before feeding $H'$ to the LLM, we first conduct a projector on $H'$ to align its dimension with the input space of the LLM. This process can be formally characterized as:

$$H^{\text{proj}} = \phi_{\text{proj}}(H'), \tag{4}$$

where $\phi_{\text{proj}}$ is implemented as a linear layer in EquiLLM.

**Geometric-aware Prompt.** One may directly input the aligned features $H^{\text{proj}}$ into the LLM to make the final predictions. However, this approach overlooks the pivotal role of the prompt, as it does not utilize the linguistic form of the prompt to effectively harness the LLM's comprehension and articulation of the specific task at hand. Therefore, in the EquiLLM framework, we carefully design task-specific prompts for different tasks to unleash domain-specific knowledge.

The prompt content for all tasks can be broadly divided into three key components: (1) *task description*, (2) *object feature description*, and (3) *object statistical information*.

▷ **Task description.** The task description consists of two parts: <Task> and <Requirement>. <Task> appears at the beginning of the prompt, providing a succinct description of the task to help the LLM quickly identify the task's objective. <Requirement> is located in the main body of the prompt and elaborates on the input-output requirements and constraints of the task, ensuring a comprehensive understanding

of the task by the LLM.

▷ **Object feature description.** The feature description of the input object begins with <Object> and primarily outlines the composition information as well as the structural characteristics of the input object.

▷ **Object statistical information.** This components starts with <Statistics>, encapsulating detailed metrics pertaining to the distribution of the object's coordinates in 3D space, including the maximum, minimum, and mean values. It is crucial to note that, unlike conventional tasks, directly incorporating absolute coordinate values into the prompt is not recommended in 3D spatial modeling tasks. This is due to the fact that transformations such as translation, reflection, or rotation applied to the input object will invariably alter the corresponding coordinate distribution, thereby violating the principle that the prompting process must remain E(3)-invariant. Consequently, we represent the coordinate distribution of the input object indirectly by computing statistical metrics related to distances. Several viable alternatives exist (*e.g.* Principal Component Analysis (PCA) on the original coordinates), as long as the computed statistical information preserves E(3)-invariance. The specific contents of the task prompts will be elucidated upon in § 3.3 and 3.4.

**Large Language Model (LLM).** After designing the prompt, we employ the tokenizer and embedding layer of the LLM to obtain the corresponding word embedding features, denoted as $P$. Subsequently, depending on the specific task, we concatenate $P$ with the invariant features $H^{\text{proj}}$ in an appropriate way. The concatenation strategies for different

tasks will be discussed in detail in § 3.3 and 3.4. Next, the concatenated features are input into the LLM with the aim of unlocking and leveraging the scientific knowledge embedded within the LLM to enhance the model's understanding and reasoning capabilities for the relevant tasks. Unlike previous works like CrystalLLM (Gruver et al., 2024) and UniST (Yuan et al., 2024) that require fine-tuning some layers within the LLM, resulting in significant computational costs and time expenditure, the EquiLLM framework freezes all parameters of the LLM, eliminating the need for additional training. This process can be roughly represented as follows:

$$[\boldsymbol{H}^{\text{llm}}, \boldsymbol{P}^{\text{llm}}] = \textbf{LLM}(\texttt{Concat}(\boldsymbol{H}^{\text{proj}}, \boldsymbol{P})), \quad (5)$$

**Equivariant Adapter.** Upon obtaining the output from the LLM, we extract the part corresponding to the invariant features, denoted as $\boldsymbol{H}^{\text{llm}}$. While directly utilizing it for final predictions may be viable for invariant tasks (*e.g.* predicting the energy of a molecular system), it is inadequate for equivariant tasks, where the core objective is to predict the 3D coordinates of objects. To address this challenge, we propose the Equivariant Adapter, which leverages one-layer EGNN (Satorras et al., 2021) to process $\boldsymbol{H}^{\text{llm}}$ while minimizing the introduction of excessive additional parameters. We select EGNN for its simple architecture and robust performance, as it has been widely adopted in prior works. Specifically, we first employ a projection layer to re-project $\boldsymbol{H}^{\text{llm}}$ back into the space corresponding to the invariant features $\boldsymbol{H}'$ and add it with $\boldsymbol{H}'$, yielding the refined feature representation $\boldsymbol{H}^r$. Then, both the equivariant coordinate features $\vec{\boldsymbol{X}}'$ from Equivariant Encoder and the refined invariant features $\boldsymbol{H}^r$ are transmitted to the EGNN, yielding the output $\vec{\boldsymbol{X}}^{\text{out}}$ and $\boldsymbol{H}^{\text{out}}$. The whole process is formally expressed as:

$$\begin{aligned} \boldsymbol{m}_{ij} &= \varphi_m\left(\boldsymbol{h}_i^r, \boldsymbol{h}_j^r, \left\|\vec{\boldsymbol{x}}_i' - \vec{\boldsymbol{x}}_j'\right\|\right), \\ \boldsymbol{h}_i^{\text{out}} &= \boldsymbol{h}_i^r + \varphi_h\left(\boldsymbol{h}_i^r, \sum_{j \in \mathcal{N}(i)} \boldsymbol{m}_{ij}\right), \quad (6) \\ \vec{\boldsymbol{x}}_i^{\text{out}} &= \vec{\boldsymbol{x}}_i' + \frac{1}{|\mathcal{N}(i)|}\sum_{j \in \mathcal{N}(i)} \varphi_x\left(\boldsymbol{m}_{ij}\right) \cdot \left(\vec{\boldsymbol{x}}_i' - \vec{\boldsymbol{x}}_j'\right), \end{aligned}$$

where $\varphi_m$, $\varphi_x$, and $\varphi_h$ denote Multi-Layer Perceptrons (MLPs), and $\mathcal{N}(i)$ refers to the set of neighboring nodes associated with the $i$-th node. Specifically, $\boldsymbol{m}_{ij}$ represents an E(3)-invariant message transmitted from node $j$ to node $i$, which is utilized to aggregate and refine the feature vector $\boldsymbol{h}_i^r$ via the function $\varphi_h$. Regarding the update of $\vec{\boldsymbol{x}}_i'$, the function $\varphi_x$ is employed to compute a scalar $\varphi_x(\boldsymbol{m}_{ij})$, which is subsequently multiplied by the difference $\vec{\boldsymbol{x}}_i' - \vec{\boldsymbol{x}}_j'$ to retain directional information, while incorporating residual connections to ensure translation equivariance.

**Comparison with LLaVA.** Our method does not simply replace LLaVA's encoder with an equivariant GNN encoder, as that would compromise the framework's overall equivariance. Instead, EquiLLM introduces an innovative design, as

shown in Fig. 1. First, the equivariant GNN encoder extracts both equivariant and invariant features, but only the invariant features are fed into the LLM, unlike LLaVA where the LLM receives all encoder outputs. Then, after LLM processing, the output is concatenated with the encoder's equivariant features via a skip connection and passed to the equivariant adapter module to generate both equivariant and invariant predictions.

Our EquiLLM framework guarantees that the overall architecture preserves the critical property of E(3)-equivariance in 3D space while also avoiding the introduction of lengthy text context due to direct 3D coordinate input, which could severely affect the efficiency of training and inference. The rigorous mathematical proof of the framework's equivariance properties can be found in Appendix A. Moreover, compared with domain-specific Equivariant Encoder, EquiLLM introduces only two projection layers and a one-layer EGNN network, significantly reducing the additional training parameters in comparison to the existing literature (Jin et al., 2024; Yuan et al., 2024). Finally, our EquiLLM framework demonstrates exceptional flexibility and can be applied to various geometric modeling tasks, showcasing its robustness and generalizability.

### 3.3 Applications on Dynamic Simulation

In this section, we will present a detailed discussion on the application of our EquiLLM in dynamic simulation.

While our model is applicable to the simulations of both molecular dynamics and human motions, we only illustrate molecular dynamics here and provide details on human motions in Appendix D. Given 3D coordinate trajectory of a physical system (*e.g.,* molecules) over $T$ frames $\vec{\boldsymbol{X}} \in \mathbb{R}^{T \times N \times 3}$, along with the invariant features $\boldsymbol{H} \in \mathbb{R}^{N \times c_a}$ encoded by atomic numbers, the model aims to infer future trajectories $\vec{\boldsymbol{X}} \in \mathbb{R}^{F \times N \times 3}$ for $F$ subsequent frames.

**Geometric-aware Prompt.** Here, we will provide a general overview of the contents encompassed within the prompt, with a more thorough exposition available in Appendix E.1.

▷ **Task description.** The model is tasked with predicting the 3D coordinates $(x, y, z)$ of heavy atoms for next $F$ frames based on the information from the previous $T$ frames.

▷ **Object feature description.** For molecular systems, the emphasis is on compositional information and structural characteristics.

▷ **Object statistical information.** Given that the intrinsic properties of each element remain invariant throughout temporal evolution, performing frame-wise computation of coordinate statistics to construct the prompt and then concatenating prompt features with invariant features at the node level would result in redundant information and in-

troduce unnecessary computational complexity. Hence, we aggregate statistics across all $T$ frames for all $N$ elements. Specifically, we compute the centroid $\bar{x} = \sum_i^{N_a} \sum_t^T \vec{x}_{t,i}$ from $T \times N$ 3D coordinates, followed by calculating the maximum, minimum, and mean distances from all $T \times N_a$ coordinates to this central point. The final representation concatenates $P$ and $\boldsymbol{H}^{\mathrm{proj}}$ along the temporal dimension.

**Task Formulation and Training Objective.** With $\mathcal{G}_{1:T} := \{\mathcal{G}_t = (\vec{\boldsymbol{X}}_t, \boldsymbol{H}_t, \mathcal{E})\}_{t=1}^T$ and $\mathcal{G}_{T+1:T+F} := \{\mathcal{G}_t = (\vec{\boldsymbol{X}}_t, \boldsymbol{H}_t, \mathcal{E})\}_{t=T+1}^{T+F}$, we provide the entire process as follows:

$$\mathcal{G}_{T+1:T+F} = \phi(\mathcal{G}_{1:T}, \boldsymbol{P}). \tag{7}$$

Let $\vec{\boldsymbol{X}}_{T+f}^{\mathrm{gt}}$ denote the ground-truth 3D coordinates for the time period from $T+1$ to $T+F$, we define the object function as $\mathcal{L} = \frac{1}{|F|} \sum_{f=1}^F \ell_{\mathrm{mse}}(\vec{\boldsymbol{X}}_{T+f}^{\mathrm{out}}, \vec{\boldsymbol{X}}_{T+f}^{\mathrm{gt}})$ refers to the mean squared error (MSE).

### 3.4 Applications on Antibody Design

In this section, we will present a detailed discussion on the application of our EquiLLM in antibody design.

Antibodies are Y-shaped proteins primarily responsible for recognizing and binding to specific antigens. Current research predominantly focuses on the variable region. The variable region is present in both the heavy and light chains of the antibody and can be further subdivided into the framework region and three Complementarity-Determining Regions (CDRs). These six CDRs are critical in determining the affinity between the antibody and antigen, with the CDR-H3 region on the heavy chain exhibiting the most pronounced variability. Consequently, the primary objective of this paper is to predict the amino acid sequence and the 3D coordinates of the CDR-H3 region, given the antibody-antigen complexes excluding the CDR-H3 region. In antibody design task, each node in $\mathcal{V}$ associates with a trainable feature $\boldsymbol{h}_i \in \mathbb{R}^{c_r}$ encoded by amino acid type and a matrix of 3D coordinates $\vec{\boldsymbol{Z}}_i \in \mathbb{R}^{4 \times 3}$. We choose 4 backbone atoms $\{\mathrm{N}, \mathrm{C}_\alpha, \mathrm{C}, \mathrm{O}\}$ to constitute $\vec{\boldsymbol{Z}}_i$.

**Geometric-aware Prompt.** Here, we will provide a general overview of the contents encompassed within the prompt, with a more thorough exposition to be presented in Appendix E.2.

▷ **Task description.** The model is tasked with predicting both the 1D sequence and 3D coordinates of CDR-H3 region.

▷ **Object feature description.** The structural features of the light chain, heavy chain, and antigen, which are described individually.

▷ **Object statistical information.** In contrast to tasks involving dynamic simulation, where each physical object comprises only dozens of elements, the antibody-antigen complex consists of three chains containing hundreds of amino acids. Therefore, computing the statistical information for each amino acid individually may exceed the input token limit. To address this, we compute $\mathrm{C}_\alpha$ atom statistics at both the chain-level and residue-level. At the chain-level, we first calculate the centroid coordinates for the light chain, heavy chain, and antigen chain and compute the distances between pairs of chains. Additionally, we calculate the distance between the two most distant amino acids within each chain, thereby constructing the corresponding prompt $\boldsymbol{P}_c$. At the residue-level, we calculate the distance from each amino acid to the centroid of its respective chain and compute the maximum, minimum, and mean distances, thus constructing the prompt $\boldsymbol{P}_r$. Finally, we concatenate $\boldsymbol{P}_c$, $\boldsymbol{P}_r$, and $\boldsymbol{H}^{\mathrm{proj}}$ along the amino acid sequence dimension.

**Task Formulation and Training Objective.** With $\mathcal{G} = (\vec{\boldsymbol{X}}, \boldsymbol{H}, \mathcal{E})$, where $(\vec{\boldsymbol{X}}, \boldsymbol{H}) := \{(\vec{\boldsymbol{Z}}_i, \boldsymbol{h}_i)\}_{i=1}^N$, the entire process is delineated as follows:

$$\begin{aligned} \boldsymbol{H}^{\mathrm{out}}, \vec{\boldsymbol{X}}^{\mathrm{out}} &= \phi_r(\mathcal{G}, \boldsymbol{P}_r), \\ \boldsymbol{y}^{\mathrm{out}} &= \mathtt{Softmax}(\boldsymbol{H}^{\mathrm{out}}), \end{aligned} \tag{8}$$

where $N$ denotes the number of residues in CDR-H3 region; $\boldsymbol{y}^{\mathrm{out}}$ and $\boldsymbol{y}^{\mathrm{gt}}$ denote the predicted distribution over all amino acid categories and the ground truth amino acid type; $\vec{\boldsymbol{X}}^{\mathrm{out}}$ and $\vec{\boldsymbol{X}}^{\mathrm{gt}}$ denote the predicted 3D structure and the ground truth 3D structure of the CDR-H3 region, respectively. The loss function is defined as $\mathcal{L} = \mathcal{L}_{\mathrm{seq}} + \lambda\mathcal{L}_{\mathrm{struct}}$, where $\mathcal{L}_{\mathrm{ce}} = \frac{1}{N}\ell_{ce}(\boldsymbol{y}^{\mathrm{out}}, \boldsymbol{y}^{\mathrm{gt}})$ denotes the cross entropy and $\mathcal{L}_{\mathrm{huber}} \frac{1}{N}\ell_{\mathrm{huber}}(\vec{\boldsymbol{X}}^{\mathrm{out}}, \vec{\boldsymbol{X}}^{\mathrm{gt}})$ denotes the Huber loss (Huber, 1992); the $\lambda$ is used to balance the two losses.

## 4 Experiments

We validate the effectiveness of the proposed EquiLLM framework on two tasks from different domains: the dynamic simulation in physics (§ 4.1 and 4.2) and the antibody design in biology (§ 4.3). Furthermore, in § 4.4, we conduct ablation studies and explore the contribution of each module. We also perform additional exploratory experiments in § 4.5.

**Datasets.** In the dynamic simulation task, to demonstrate the broad applicability of our model across varying scales, we conduct experiments on two distinct datasets: the molecular-level MD17 (Chmiela et al., 2017) dataset and the macro-level Human Motion Capture (De la Torre et al., 2009) dataset. In order to expedite the dynamics simulations, we implement a sampling strategy based on previous research (Huang et al., 2022) to extract a subset of trajectories for the purposes of training, validation, and testing. This approach involves randomly selecting an initial point and then sampling $2 \times T$ timestamps. The first $T$ timestamps are utilized as input for the models, while the remaining $T$

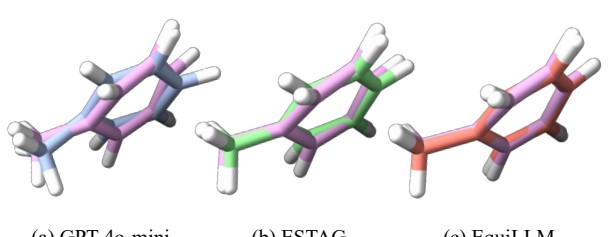

(a) GPT-4o-mini    (b) ESTAG    (c) EquiLLM

Figure 2: The visualization of the predicted structures across various methods on Toluene of the MD17 dataset, where the pink represents the ground-truth structure.

timestamps represent the future states that the models need to predict.

**Baselines and metrics.** We evaluate EquiLLM with several baseline models, including traditional GNNs: ST_GNN (Gilmer et al., 2017) and STGCN (Yu et al., 2017), equivariant GNNs: ST_TFN (Thomas et al., 2018), ST_SE(3)-Tr. (Fuchs et al., 2020), ST_EGNN (Satorras et al., 2021), and ESTAG (Wu et al., 2024). We also compare EquiLLM with existing LLMs, including GPT-4o-mini (OpenAI, 2024), Gemini-1.5-flash-latest (Team et al., 2024), and DeepSeek-V3 (Liu et al., 2024a). The models marked with "ST" are those we modified to handle multi-frame inputs by adding basic spatio-temporal aggregation, as done in (Wu et al., 2024). For evaluation, we calculate the Mean Squared Errors (MSEs) averaged across all predicted frames as the metric.

### 4.1 Molecular Dynamics

**Implementation details.** MD17 consists of time-evolving paths produced through molecular dynamics simulation for eight different small compounds (such as aspirin, benzene, and others). To ensure a fair comparison, all hyperparameters (*e.g.* learning rate, number of training epochs) are kept consistent across our model and all other baselines. Detailed information can be found in Appendix B.1. We utilize the ESTAG (Wu et al., 2024) as the Equivariant Encoder and GPT-2 (Radford et al., 2019) [1] as the language model within our EquiLLM framework. ESTAG utilizes Equivariant Discrete Fourier Transform to extract periodic patterns, and further employs alternating equivariant spatial and temporal modules to enhance the modeling of physical dynamics. Unless otherwise specified, in all experiments presented in this paper, the LLM module in EquiLLM remains frozen, while all other modules are learnable and trained from scratch.

**Results.** Table 1 presents the performance of all models on MD17 dataset under the setting of predicting 10 frames

---

[1] A more powerful LLM may lead to a superior performance. Here we use GPT-2 for concept validation.

from an input of 10 frames. EquiLLM (DST) and EquiLLM (PCA) represent the utilization of distance-related and principal component analysis-related information as object statistical information, respectively. From the table, the following conclusions can be drawn: **1.** The proposed EquiLLM framework achieves state-of-the-art (SOTA) performance on all eight molecules, demonstrating its superiority; **2.** Compared to our Equivariant Encoder model ESTAG, EquiLLM achieves a performance improvement of 5.41% to 42.76% on six molecules, indicating that EquiLLM effectively leverages knowledge from LLMs to enhance the prediction of molecular dynamics trajectories; **3.** We also tested the prediction capability of several leading LLMs, including GPT-4o-mini, Gemini-1.5-flash-latest, and DeepSeek-V3. Using the same prompt as EquiLLM, we provide the 3D coordinates of all atoms from the past 10 frames to these LLMs, allowing it to predict the coordinates of all atoms in the following 10 frames. The result shows that these LLMs significantly underperforms most baseline methods in prediction accuracy, indicating its weaker capability in directly predicting 3D coordinates. In contrast, our EquiLLM framework, by providing structured molecular descriptions and statistical constraints, enables the LLM to combine its pretrained knowledge with the specific task, thus significantly reduce the predicted MSE; **4.** We further test pretrained models with canonicalization. On the MD17 dataset, we first subtract the mean from coordinates to ensure translational invariance, then perform SVD decomposition for rotational invariance. We directly feed this canonicalized data into GPT-4o-mini, with results shown in Row 8 of Table 1. The results demonstrate that while canonicalization indeed improves model's predictive capability, there remains a remarkable performance gap compared to our EquiLLM. This suggests that direct prediction of 3D coordinates remains suboptimal for current LLMs.

### 4.2 Human Motion Simulation

**Implementation details.** The Human Motion Capture dataset contains human motion trajectory data across multiple scenes. We focus primarily on two sub-datasets: Subject #35 (Walk) and Subject #102 (Basketball). To ensure a fair comparison, all hyperparameters (*e.g.* learning rate, number of training epochs) are kept consistent across our model and all other baselines. Detailed information is provided in Appendix B.2. We utilize the ESTAG as the Equivariant Encoder and GPT-2 as the language model within our EquiLLM framework.

**Results.** Table 2 presents a performance comparison of all models on the Walk and Basketball datasets under settings requiring the prediction of 10, 15, and 20 frames, respectively. From the table, it is evident that EquiLLM framework achieves SOTA performance across all six settings, with a performance improvement ranging from 5.63%

Table 1: Predicted MSE ($\times 10^{-3}$) on MD17 dataset.

| | Aspirin | Benzene | Ethanol | Malonaldehyde | Naphthalene | Salicylic | Toluene | Uracil |
|---|---|---|---|---|---|---|---|---|
| ST_GNN | $7.180_{\pm 0.003}$ | $1.359_{\pm 0.001}$ | $2.108_{\pm 0.001}$ | $5.620_{\pm 0.018}$ | $2.397_{\pm 0.017}$ | $2.646_{\pm 0.003}$ | $2.233_{\pm 0.011}$ | $1.913_{\pm 0.012}$ |
| ST_TFN | $7.389_{\pm 0.139}$ | $1.849_{\pm 0.003}$ | $2.041_{\pm 0.001}$ | $5.346_{\pm 0.006}$ | $3.555_{\pm 0.110}$ | $5.728_{\pm 0.015}$ | $2.979_{\pm 0.167}$ | $4.272_{\pm 0.035}$ |
| STGCN | $21.08_{\pm 0.001}$ | $654.7_{\pm 0.001}$ | $7.102_{\pm 0.001}$ | $32.87_{\pm 0.001}$ | $5.421_{\pm 0.001}$ | $3.501_{\pm 0.001}$ | $3.679_{\pm 0.001}$ | $7.142_{\pm 0.001}$ |
| ST_SE(3)-Tr. | $6.234_{\pm 0.019}$ | $1.835_{\pm 0.001}$ | $1.765_{\pm 0.001}$ | $5.277_{\pm 0.070}$ | $3.256_{\pm 0.018}$ | $4.737_{\pm 0.016}$ | $2.104_{\pm 0.011}$ | $3.900_{\pm 0.006}$ |
| ST_EGNN | $6.682_{\pm 0.380}$ | $1.482_{\pm 0.161}$ | $2.145_{\pm 0.001}$ | $4.729_{\pm 0.029}$ | $4.034_{\pm 0.028}$ | $6.296_{\pm 0.157}$ | $2.881_{\pm 0.002}$ | $3.394_{\pm 0.267}$ |
| Equiformer | 10.130 | 2.000 | 1.880 | 8.050 | 3.430 | 5.790 | 2.090 | 4.380 |
| GPT-4o-mini | 13.070 | 9.581 | 5.011 | 9.910 | 35.155 | 10.627 | 8.132 | 9.762 |
| GPT-4o-mini + canonicalization | 11.783 | 3.055 | 4.512 | 8.916 | 8.263 | 9.751 | 6.364 | 8.989 |
| Gemini-1.5-flash-latest | 17.347 | 7.586 | 8.871 | 15.495 | 15.188 | 17.978 | 15.426 | 16.935 |
| DeepSeek-V3 | 12.009 | 3.648 | 6.729 | 10.247 | 7.883 | 10.423 | 6.941 | 9.428 |
| ESTAG | $3.263_{\pm 0.065}$ | $0.891_{\pm 0.083}$ | $1.090_{\pm 0.001}$ | $2.046_{\pm 0.085}$ | $2.036_{\pm 0.350}$ | $3.134_{\pm 0.094}$ | $1.634_{\pm 0.149}$ | $1.852_{\pm 0.066}$ |
| EquiLLM (DST) | $\underline{2.391}_{\pm 0.233}$ | $\underline{0.732}_{\pm 0.058}$ | $\mathbf{1.031}_{\pm 0.001}$ | $\underline{1.671}_{\pm 0.025}$ | $\mathbf{1.453}_{\pm 0.071}$ | $\underline{2.162}_{\pm 0.137}$ | $\underline{1.178}_{\pm 0.186}$ | $\underline{1.060}_{\pm 0.194}$ |
| EquiLLM (PCA) | $\mathbf{1.931}_{\pm 0.084}$ | $\mathbf{0.552}_{\pm 0.075}$ | $\underline{1.044}_{\pm 0.001}$ | $\mathbf{1.634}_{\pm 0.024}$ | $\underline{1.454}_{\pm 0.141}$ | $\mathbf{1.852}_{\pm 0.079}$ | $\mathbf{1.049}_{\pm 0.126}$ | $\mathbf{0.940}_{\pm 0.082}$ |

Table 2: Predicted MSE ($\times 10^{-2}$) on Motion dataset.

| Method | Walk | | | Basketball | | |
|---|---|---|---|---|---|---|
| | R=10 | R=15 | R=20 | R=10 | R=15 | R=20 |
| ST_GNN | $1.150_{\pm 0.001}$ | $2.544_{\pm 0.814}$ | $2.765_{\pm 0.032}$ | $34.536_{\pm 0.747}$ | $133.731_{\pm 11.677}$ | $278.246_{\pm 2.225}$ |
| ST_TFN | $9.584_{\pm 0.156}$ | $20.667_{\pm 1.533}$ | $31.437_{\pm 0.120}$ | $168.674_{\pm 0.556}$ | $358.881_{\pm 1.661}$ | $613.755_{\pm 3.256}$ |
| ST_GCN | $18.737_{\pm 1.351}$ | $19.467_{\pm 0.577}$ | $20.498_{\pm 2.232}$ | $275.744_{\pm 13.322}$ | $516.462_{\pm 91.545}$ | $662.488_{\pm 21.859}$ |
| ST_SE(3)-Tr. | $5.248_{\pm 0.132}$ | $10.869_{\pm 0.596}$ | $20.999_{\pm 0.156}$ | $178.677_{\pm 4.022}$ | $390.518_{\pm 3.260}$ | $621.004_{\pm 12.186}$ |
| ST_EGNN | $2.867_{\pm 0.011}$ | $4.189_{\pm 0.172}$ | $8.644_{\pm 1.620}$ | $30.813_{\pm 0.122}$ | $72.963_{\pm 1.295}$ | $152.551_{\pm 1.466}$ |
| ESTAG | $0.709_{\pm 0.052}$ | $1.877_{\pm 0.211}$ | $3.464_{\pm 1.127}$ | $10.507_{\pm 0.073}$ | $33.636_{\pm 0.425}$ | $76.548_{\pm 0.916}$ |
| EquiLLM | $\mathbf{0.539}_{\pm 0.011}$ | $\mathbf{1.300}_{\pm 0.134}$ | $\mathbf{2.213}_{\pm 0.160}$ | $\mathbf{9.438}_{\pm 0.202}$ | $\mathbf{30.371}_{\pm 0.068}$ | $\mathbf{72.233}_{\pm 0.954}$ |

Table 3: Results on RAbD benchmark.

| Method | AAR ↑ | TM-score ↑ | RMSD ↓ |
|---|---|---|---|
| RosettaAD | 22.50% | 0.9435 | 5.52 |
| LSTM | 22.36% | - | - |
| C-LSTM | 22.18% | - | - |
| RefineGNN | 29.79% | 0.8303 | 7.55 |
| C-RefineGNN | 28.90% | 0.8317 | 7.21 |
| GeoAB | 36.43% | **0.9836** | 1.79 |
| MEAN | 36.77% | 0.9812 | 1.81 |
| EquiLLM | **38.97 %** | 0.9830 | **1.73** |

Table 4: Ablation studies ($\times 10^{-3}$) on MD17 dataset.

| | Aspirin | Benzene | Ethanol | Malonaldehyde | Naphthalene | Salicylic | Toluene | Uracil |
|---|---|---|---|---|---|---|---|---|
| EE (ESTAG) | 3.263 | 0.891 | 1.090 | 2.046 | 2.036 | 3.134 | 1.634 | 1.852 |
| LLM (w/o Prompt) + EE | 2.524 | 0.810 | 1.062 | 2.326 | 2.363 | 2.180 | 1.538 | 1.425 |
| LLM + EE | 3.083 | 0.773 | 1.128 | 2.100 | 2.899 | 2.581 | 1.756 | 1.418 |
| w/o Prompt | 3.671 | 0.860 | 1.092 | 2.479 | 2.837 | 2.193 | 1.941 | 1.542 |
| w/o Object Feature | 3.122 | 0.833 | 1.080 | 2.255 | 2.297 | 2.470 | 1.627 | 1.387 |
| w/o Statistics | 3.532 | 0.820 | 1.054 | 1.889 | 2.286 | 2.528 | 1.650 | 1.463 |
| EquiLLM | **2.391** | **0.732** | **1.031** | **1.671** | **1.453** | **2.162** | **1.178** | **1.060** |

to 36.11%. This demonstrates that EquiLLM effectively handles predictions over varying prediction lengths, exhibiting excellent robustness and generalization ability

### 4.3 Antibody Design

Following previous study MEAN (Kong et al., 2022), we selected complete antibody-antigen complexes from the SAbDab (Dunbar et al., 2014) dataset to construct the training and validation sets. First, we performed clustering based on CDRs, grouping complexes with CDR sequence identity above 40% into the same cluster. Then, the training and validation sets were partitioned in the same manner as in

MEAN. For test set, we selected 60 diverse complexes from the RAbD (Adolf-Bryfogle et al., 2018) dataset to evaluate the performance of different methods. Before starting the experiments, we remove samples from the training and validation sets that belong to the same cluster as the test set to prevent data leakage. More experimental results can be found in Appendix C.3.

**Baselines and metrics.** We compared our EquiLLM with seven methods, including RosettaAD (Adolf-Bryfogle et al., 2018), LSTM (Saka et al., 2021; Akbar et al., 2022), RefineGNN (Jin et al., 2022), MEAN (Kong et al., 2022), GeoAB (Lin et al., 2024), and two variants of LSTM and

RefineGNN, C-LSTM and C-RefineGNN, which utilize the full contextual information. We use AAR and RMSD to reflect the recovery ratio of the CDR-H3 amino acid sequence and the accuracy of the corresponding 3D structure prediction. Additionally, we employ the TM-score (Zhang & Skolnick, 2004; Xu & Zhang, 2010) to measure the global similarity between two protein structures. We utilize the MEAN (Kong et al., 2022) as the Equivariant Encoder and GPT-2 as the language model within our framework. Detailed information can be found in Appendix B.3.

**Results.** Table 3 presents the performance of all models on the RAbD dataset. It can be concluded from the table: **1.** The proposed EquiLLM framework achieves the best performance in both AAR and RMSD metrics, with comparable results in the TM-score metric. The SOTA method GeoAB achieves superior performance in TM-score by incorporating more detailed geometric constraints, such as bond lengths, bond angles, and torsion angles, into the model, which enhances its overall structural prediction capabilities; **2.** Compared to the Equivariant Encoder model MEAN, EquiLLM shows significant improvement across all metrics, demonstrating that EquiLLM successfully leverages LLM's knowledge integration and constraint-handling abilities while effectively utilizing LLM's capacity for understanding and reasoning 1D sequences.

### 4.4 Ablation Studies

In this section, we delve into the design of the EquiLLM framework, analyzing the impact of different architectural designs and prompt configurations on model performance. The experimental results are shown in Table 4, where EE represents the Equivariant Encoder. More ablation studies can be found in Appendix C.2.

**Architecture Design. 1.** The results in the second row indicate that processing raw features through the LLM before feeding them into the Equivariant Encoder, while omitting the Equivariant Adapter for a simpler architecture, yields a performance improvement. This finding validates that the LLM has a fundamental capability to process and integrate structured information effectively. **2.** However, to fully exploit the potential of the LLM model, it is necessary to leverage prompts to capitalize on its strengths in text understanding. Building upon the second-row model, we perform experiments by adding prompts, as the results shown in the third row. The results indicate a significant performance drop. We speculate that this is due to the large semantic space difference between the unprocessed raw feature $H$ and the text features, which hampers the model's prediction capabilities. This suggests that LLM requires an appropriate interface to harness its advantages. This led to the design of the current EquiLLM framework.

**Prompt Design. 3.** From Table 4, it is evident that either completely removing prompt or reducing its content leads to a decline in performance. This observation reinforces our design philosophy: LLMs necessitate comprehensive information, including molecular descriptions and statistical constraints, to fully utilize their knowledge integration and constraint reasoning capabilities. This, in turn, facilitates more accurate predictive guidance.

### 4.5 Further Exploratory Investigations

In this section, we conduct additional exploratory experiments to further investigate the EquiLLM framework. More exploratory investigations can be found in Appendix C.1.

Following CrystalLLM, we finetune Llama-7b (Touvron et al., 2023) on MD17. Due to token length limitations in our prediction task (predicting 10 frames), we select the smallest molecule, Ethanol (3 heavy atoms) for evaluation. We conduct three settings: (1) 500 samples (original setup in main experiments) trained for 10 epochs; (2) 30,000 samples trained for 1 epoch; (3) 30,000 samples trained for 1 epoch with canonicalization. The results in Table 5 reveal that without canonicalization, 500-sample and 30,000-sample fine-tuned models perform poorly, lagging behind EquiLLM by two orders of magnitude. Remarkably, when we incorporate canonicalization, the model's predictive performance improved by a factor of 100, even surpassing GPT-4o-mini. This compelling result demonstrates that the combination of canonicalization with direct LLM fine-tuning is indeed promising and warrants further investigation.

Table 5: Results of fine-tuning Llama on MD17.

|         | Setting 1 | Setting 2 | Setting 3 | EquiLLM |
|---------|-----------|-----------|-----------|---------|
| Ethanol | 460       | 457       | 4.446     | 1.031   |

## 5 Conclusion

We present EquiLLM, a framework that synergizes the strengths of LLMs and geometric GNNs to address the dual challenges of $E(3)$-equivariance and knowledge integration in 3D physical system modeling. By introducing geometry-aware prompting and a modular architecture that isolates invariant and equivariant processing, EquiLLM circumvents inherent limitations of LLMs in spatial reasoning while enabling the infusion of domain-specific knowledge through flexible prompting strategies. The separation of roles—LLMs as invariant feature processors and geometric GNNs as directional information handlers—provides a principled approach to preserving symmetry constraints. In future work, we plan to explore optimal prompting strategies for better leveraging domain knowledge and extending this framework to broader scientific tasks. We hope the EquiLLM framework will serve as a valuable reference for applying LLMs in scientific domains.

## Acknowledgements

This work was jointly supported by the following projects: the National Natural Science Foundation of China (No. 62376276); Beijing Nova Program (No. 20230484278); the Fundamental Research Funds for the Central Universities, and the Research Funds of Renmin University of China (23XNKJ19); Damo Academy (Hupan Laboratory) through Damo Academy (Hupan Laboratory) Innovative Research Program.

## Impact Statement

This paper presents work whose goal is to advance the field of Machine Learning. There are many potential societal consequences of our work, none which we feel must be specifically highlighted here.

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

# A    Proof of EquiLLM's Equivariance

We now present the theoretical foundations for the symmetries exhibited by EquiLLM. Specifically, we analyze two key properties: the invariance of the node representation $h_i^{\text{out}}$ and the equivariance of the node coordinate $\vec{x}_i^{\text{out}}$. To establish these properties, we break down the proof into two components, demonstrating the equivariance of both the encoder and the adapter.

*Proof.*   Consider the rotation/reflection matrix $O \in \mathbb{R}^{3 \times 3}$ and the translation vector $b \in \mathbb{R}^{3 \times 1}$.

**Equivariance of the Encoder.**   The equivariance of the encoder can be expressed as:

$$g \cdot \mathcal{G}' = \phi_e(g \cdot \mathcal{G}^{\text{in}}), \tag{9}$$

where $g$ is an element of the E(3) group. The proof of this property is detailed in the appendix of the original ESTAG paper (Wu et al., 2024). In essence, this ensures that the output node features $H'$ are invariant under transformations, while the output coordinates $\vec{X}'$ are equivariant.

**Equivariance of the Adapter.**   Next, we prove the equivariance of the adapter. Recall that the node features from the encoder are first processed through a projector and then passed through a large language model, both of which are invariant operations. Consequently, the resulting node features $H^r$ remain invariant, while the coordinates $\vec{X}'$ remain equivariant. These outputs serve as inputs to the adapter, and we analyze their transformation properties below:

(1) Invariance of pairwise interactions. The interaction term $m_{ij}$ between nodes $i$ and $j$ is defined as:

$$
\begin{aligned}
\tilde{m}_{ij} &= \varphi_m \left( h_i^r, \, h_j^r, \, \left\| (O\vec{x}_i' + b) - (O\vec{x}_j' + b) \right\| \right) \\
&= \varphi_m \left( h_i^r, \, h_j^r, \, \left\| \vec{x}_i' - \vec{x}_j' \right\| \right) \\
&= m_{ij},
\end{aligned}
$$

where, $\tilde{m}_{ij}$ denotes the transformed variable. Since the Euclidean distance is invariant under rotations and translations, $m_{ij}$ remains unchanged.

(2) Invariance of output node features $h_i^{\text{out}}$. Its update rule is given by:

$$h_i^{\text{out}} = h_i^r + \varphi_h \left( h_i^r, \sum_{j \in \mathcal{N}(i)} m_{ij} \right).$$

Here, both $h_i^r$ and $m_{ij}$ are invariant variables, and the multi-layer perceptron (MLP) $\phi_h$ and summation $\sum_{j \in \mathcal{N}(i)}$ are invariant operations. Therefore, it follows that $h_i^{\text{out}}$ is also invariant.

(3) Equivariance of output coordinates $\vec{x}_i^{\text{out}}$. The transformation is formulated as:

$$
\begin{aligned}
\tilde{\vec{x}}_i^{\text{out}} &= O\vec{x}_i' + b + \frac{1}{|\mathcal{N}(i)|} \sum_{j \in \mathcal{N}(i)} \varphi_x(m_{ij}) \cdot \left( (O\vec{x}_i' + b) - (O\vec{x}_j' + b) \right) \\
&= O\vec{x}_i' + b + \frac{1}{|\mathcal{N}(i)|} \cdot O \cdot \sum_{j \in \mathcal{N}(i)} \varphi_x(m_{ij}) \cdot \left( \vec{x}_i' - \vec{x}_j' \right) \\
&= O \left( \vec{x}_i' + \frac{1}{|\mathcal{N}(i)|} \sum_{j \in \mathcal{N}(i)} \varphi_x(m_{ij}) \cdot \left( \vec{x}_i' - \vec{x}_j' \right) \right) + b \\
&= O \cdot \vec{x}_i^{\text{out}} + b.
\end{aligned}
$$

Thus, the output coordinates transform consistently with the input coordinates under rotations and translations, confirming their equivariance.

**Conclusion.**   In summary, we have rigorously demonstrated that the node representations $h_i^{\text{out}}$ are invariant, while the node coordinates $\vec{x}_i^{\text{out}}$ are equivariant under E(3) transformations. These results align with our initial claims about the symmetries of EquiLLM.    $\square$

# B    Dataset Details

## B.1    Implementation Details on MD17 dataset

The first column in Table 6 delineates the standardized parameter configuration utilized throughout our experiments on the MD17 dataset. This uniform parameterization scheme has been implemented across both our EquiLLM architecture and all other baseline methodologies. All computational experiments, encompassing model training, validation and testing phases, are executed on a single NVIDIA A100-80G GPU. The training, validation, and testing sets consist of 500, 2000, and 2000 samples, respectively.

The MD17 dataset exhibits a characteristic graph structure with a maximum cardinality of 13 nodes. Regarding graph topology construction, we establish interatomic connectivity through a distance-based criterion, where atomic pairs separated by less than the threshold parameter $\lambda$ are classified as primary neighbors. The graph connectivity framework extends to incorporate both primary and secondary neighboring relationships, thereby establishing a comprehensive edge structure to support subsequent message passing.

Table 6: Hyper-parameters of EquiLLM and other methods. The previous length $T_p$ denotes the length of input sequence, the future length $T_f$ denotes the length of output sequence, the time lag $\Delta t$ denotes the interval between two timestamps, the hidden size denotes the size of hidden states in all Multi-Layer Perceptrons (MLPs) within the EquiLLM framework, and the layer denotes the number of layers.

| Hyper-parameter | MD17 | Motion Capture |
|---|---|---|
| Learning Rate | 5e-3 | 5e-3 |
| Epochs | 500 | 500 |
| Previous Length $T_p$ | 10 | 10 |
| Future Length $T_f$ | 10 | 10, 15, 20 |
| Time Lag $\Delta t$ | 10 | 1 |
| Hidden Size | 16 | 16 |
| Layer | 2 | 2 |

## B.2    Implementation Details on Motion Capture

The second column in Table 6 outlines the standardized configuration utilized for Human Motion Capture dataset evaluations. This consistent parameterization is implemented across our EquiLLM architecture and all other baselines. All computational experiments, encompassing model training, validation and testing phases, are executed on a single NVIDIA A100-80G GPU. We maintain the experimental configurations and dataset partitions specified in (Wu et al., 2024). For subject #35 (Walk), the dataset comprises 1100 training, 600 validation, and 600 testing trajectories, whereas subject #102 (Basketball) includes 600 training, 300 validation, and 300 testing trajectories.

The Motion Capture dataset comprises graphs with a maximum of 31 nodes per instance. Graph connectivity is established by defining directly adjacent joints as primary neighbors. Each joint forms edges with both primary and secondary adjacent nodes to enable efficient subsequent message passing. All joints share identical invariant feature representations, initialized as unit vectors.

## B.3    Implementation Details on Antibody Design

We adhere to the experimental configurations established in MEAN (Kong et al., 2022) to ensure fair comparisons. Specifically, EquiLLM maintains identical hyper-parameters with MEAN: a 64-dimensional trainable embedding for each amino acid type, 128-dimensional hidden states, 3 network layers, a batch size of 16, and 20 training epochs. The Adam optimizer is employed with an initial learning rate of 0.001, which decays by 5% per epoch. We utilize the same dataset splits as MEAN, excluding one antigen-antibody complex from the training set due to missing light chain data. For GeoAB (Lin et al., 2024), we reproduce the results using the official implementation on our datasets.

To ensure fair comparisons, we adopt the same settings as MEAN (Kong et al., 2022). Specifically, EquiLLLM shares the same hyperparameters as MEAN: the trainable embedding size for each amino acid type is 64; the hidden state size is 128; the number of layers is 3; the batch size is 16; and the number of training epochs is 20. We employ the Adam optimizer

with a learning rate $lr$ of 0.001, which is decayed by a factor of 0.95 after each epoch. Furthermore, we use the same training, validation, and test datasets as those in MEAN, except for one antigen-antibody complex in the training set that was excluded due to the absence of the light chain. For the GeoAB (Lin et al., 2024) method, we reproduce the results on our datasets using the official implementation.

## C More Experiment Results

### C.1 More Exploratory Experiments

**Fine-tuning the LLM Layers.** We set the LLM's parameters to be trainable and fine-tune the model on the SAbDab dataset. However, the experimental results ( Table 7, first row) show performance degradation, suggesting that fine-tuning may compromise the original information encoded in the LLM, particularly since the dataset used for fine-tuning is not large enough.

Table 7: EquiLLM with different backbones.

| Method | AAR | TM-score | RMSD |
| --- | --- | --- | --- |
| Finetune LLM | 38.57% | 0.9819 | 1.77 |
| Normal GNN encoder | 32.32% | 0.9308 | 4.14 |
| Qwen2.5-3B | **39.04%** | 0.9828 | 1.76 |
| Original | 38.97% | **0.9830** | **1.73** |

**Normal GNN Encoder.** We additionally replace the original equivariant GNN encoder with a normal GNN encoder. As shown in Row 2 of Table 7, the model exhibits significant performance degradation, demonstrating the importance of maintaining E(3)-equivariance when modeling 3D structures. Furthermore, in our original equivariant GNN encoder, equivariant and invariant features interact through message passing and feature updating, with 3D spatial distances explicitly encoded. As established in Section 3.3 of PAINN (Schütt et al., 2021), incorporating distance information across stacked layers implicitly models angular relationships, enabling the output invariant features to inherently capture spatial geometric information.

**Other LLM Backbone.** We conduct additional evaluations using the Qwen2.5-3B model (see Table 7, Row 3). Although it shows a marginal improvement in AAR, we observe slight decreases in RMSD and TM-score performance. We hypothesize that the language model's capability remains constrained by limited text-3D structure paired data; otherwise, upgrading the LLM component could yield significant gains. We leave this exploration for future work.

### C.2 More Ablation Studies

Table 8: Ablation studies on RAbD dataset.

| Method | AAR | TM-score | RMSD |
| --- | --- | --- | --- |
| w/o object feature | 38.32% | 0.9826 | 1.76 |
| w/o LLM | 37.58% | 0.9818 | 1.79 |
| w/o prompt1 | 37.84% | 0.9820 | 1.76 |
| w/o prompt2 | 38.57% | 0.9823 | 1.77 |
| w/o prompt3 | 38.52% | 0.9827 | 1.74 |
| EquiLLM | **38.97%** | **0.9830** | **1.73** |

We further conduct more ablations on the RAbD dataset. The experimental results are shown in Table 8.

**Role of External Knowledge and LLM.** We indirectly demonstrate, through ablation experiments, that the model's performance suffers without properly designed prompts to activate the LLM's knowledge. Specifically, when removing antigen, light chain, and heavy chain feature descriptions from antibody design prompts (Table 8, Row 1), we observe performance

degradation, highlighting how domain-specific knowledge enhances EquiLLM's geometric modeling capabilities. Moreover, We also remove the LLM module, with results presented in Row 2 of Table 8. The model exhibits significant performance degradation, underscoring the critical role of LLM in our framework.

**Impact of Prompt Components.** We conduct more detailed prompt ablations on the RAbD dataset, to investigate the impact of different prompt components on model performance. For antibody design task, the object statistical information encompasses two hierarchical levels: **1. Chain-level features:** Inter-chain centroid distances (prompt 1) and Maximum residue-residue distances within each chain(prompt 2); **2. Residue-level features:** Statistics (max/min/mean) of residue-to-centroid distances per chain(prompt 3).

As shown in Rows 3-5 of Table 8, the results demonstrate that chain-level features contribute more significantly to performance improvement compared to residue-level features. We hypothesize that this discrepancy arises because chain-level features provide macroscopic structural information that better facilitates global 3D structure understanding and modeling.

### C.3 Inference Time

As shown in Table 9 on antibody desgin task, our comparative analysis of inference times reveals that EquiLLM requires slightly more computation than state-of-the-art methods (MEAN and GeoAB), but this modest overhead is justified by its substantial accuracy gains.

Table 9: The inference time on RAbD.

|        | GeoAB  | MEAN   | EquiLLM |
|--------|--------|--------|---------|
| Time/s | 0.0265 | 0.0139 | 0.0539  |

## D    Human Motion Simulation

Given the 3D coordinate trajectory of a physical system (*e.g.,* human bodies) over $T$ frames, namely, $\vec{X} \in \mathbb{R}^{T \times N \times 3}$, along with the invariant features $H$ of all the joints (all 1s), the model aims to infer future trajectories $\vec{X} \in \mathbb{R}^{F \times N \times 3}$ for $F$ subsequent frames.

---

**Geometric-aware Prompt**

▷ **Task description.**

```
Task: Predict human basketball-related motions in 3D space.
```

▷ **Object feature description.**

```
Action: Basketball movements, such as dribbling, shooting, passing, and defense.
    Motions involve ...
```

▷ **Object statistical information.**

```
Statistics for all joints in the human body over the past 10 frames:

Distance of each joint from the origin $(0, 0, 0)$:

"Minimum distance": min_value,

"Maximum distance": max_value,

"Mean distance": mean_value.
```

---

**Task Formulation and Training Objective.**     With $\mathcal{G}_{1:T} := \{\mathcal{G}_t = (\vec{X}_t, H_t, \mathcal{E})\}_{t=1}^{T}$ and $\mathcal{G}_{T+1:T+F} := \{\mathcal{G}_t = (\vec{X}_t, H_t, \mathcal{E})\}_{t=T+1}^{T+F}$, we provide the entire process as follows:

$$\mathcal{G}_{T+1:T+F} = \phi(\mathcal{G}_{1:T}, P). \tag{10}$$

Let $\vec{X}_{T+f}^{\text{gt}}$ denote the ground-truth 3D coordinates for the time period from $T+1$ to $T+F$, we define the object function as $\mathcal{L} = \frac{1}{|F|} \sum_{f=1}^{F} \ell_{\text{mse}}(\vec{X}_{T+f}^{\text{out}}, \vec{X}_{T+f}^{\text{gt}})$ refers to the mean squared error (MSE).

# E    Detailed Geometric-aware Prompt

## E.1    Molecular Dynamics Simulation

In EquiLLM (DST), we compute the distances from all atomic 3D coordinates to the origin (0, 0, 0) after coordinate centering and derive their statistical properties. The corresponding prompt is as follows:

---

**Geometric-aware Prompt**

▷ **Task description.**

```
Task: Predict molecular 3D coordinates (x, y, z).
```

▷ **Object feature description.**

```
Molecule: Aspirin (C9H8O4) with 13 heavy atoms (9C, 4O). Structure includes
    aromatic ring with acetyl ...
```

▷ **Object statistical information.**

```
Statistics for all heavy atoms in one molecule over the past 10 frames:

Distance of each heavy atom from the origin (0, 0, 0):

"Minimum distance": min_value,

"Maximum distance": max_value,

"Mean distance": mean_value.
```

---

In EquiLLM (PCA), we employ Principal Component Analysis (PCA) to process coordinate data, project the data onto the first two principal components, and subsequently compute statistical information from the projected data. The corresponding prompt is as follows:

---

**Geometric-aware Prompt**

▷ **Task description.**

```
Task: Predict molecular 3D coordinates (x, y, z).
```

▷ **Object feature description.**

```
Molecule: Aspirin (C9H8O4) with 13 heavy atoms (9C, 4O). Structure includes
    aromatic ring with acetyl ...
```

▷ **Object statistical information.**

```
Statistics for all heavy atoms in one molecule over the past 10 frames:
- Shape of the original input tensor: original_shape
- After reshaping for PCA: new_shape

Statistics of the PCA projection of the reshaped 3D trajectory data:

- Eigenvalues of the PCA components:
  - PC1: PC1_value
  - PC2: PC2_value

- Dimension 1 (PC1):
  - Minimum: min_value
  - Maximum: max_value
  - Mean: mean_value
  - Median: median_value
  - Standard Deviation: std_value

- Dimension 2 (PC2):
```

---

```
    – Minimum: min_value
    – Maximum: max_value
    – Mean: mean_value
    – Median: median_value
    – Standard Deviation: std_value
```

## E.2   Antibody Design

**Geometric-aware Prompt**

▷ **Task description.**

Task: Predict the 3D structure and amino acid sequence of the CDR-H3 region in an antibody-antigen complex.

▷ **Object feature description.**

<Module: Antigen>

Defines binding constraints for the CDR-H3 backbone prediction ...

<Module: Heavy Chain>

Provides structural context for the CDR-H3 region ...

<Module: Light Chain>

Offers spatial context to guide the CDR-H3 ...

▷ **Object statistical information.**

Statistics of Global Features:

Distances between centroids of different chains ...:

"Heavy-Light": heavy_light_value,

"Heavy-Antigen": heavy_antigen_value,

"Light-Antigen": light_antigen_value.

Distance between the two most distant amino acids on one chain ...:

"Heavy": heavy_value,

"Light": light_value,

"Antigen": antigen_value.

Statistics of Chain-Level Features:

Distances of each amino acid's alpha carbon atom to the chain centroid's ...:

"Minimum distance": min_value,

"Maximum distance": max_value,

"Mean distance": mean_value.

