# OpenReview forum: "Large Language-Geometry Model: When LLM meets Equivariance"
_ICML.cc/2025/Conference — ICML 2025 poster_

### Official Review · Reviewer_whe6 · 2025-03-02

**Overall Recommendation:** 2

**Summary:**

EquiLLM integrates Large Language Models (LLMs) with geometric Graph Neural Networks (GNNs) to improve 3D structure and dynamics prediction. It uses an LLM for invariant feature processing, a GNN for equivariant encoding, and an adapter to ensure equivariance while leveraging external knowledge. Experiments show significant improvements in molecular dynamics, human motion, and antibody design, demonstrating strong generalizability.

**Claims And Evidence:**

See subsequent subsections.

**Essential References Not Discussed:**

No.

**Experimental Designs Or Analyses:**

See subsequent subsections.

**Methods And Evaluation Criteria:**

See subsequent subsections.

**Other Comments Or Suggestions:**

NO.

**Other Strengths And Weaknesses:**

Strengths:

1. The paper presents an innovative attempt by cleverly integrating pre-trained large language models with equivariant networks and demonstrates the feasibility of the proposed approach.
2. The paper conducts experiments on multiple tasks related to equivariant graph neural networks, proving the applicability of the method across various tasks.
3. Compared to the baseline models provided in the paper, the proposed method demonstrates superior performance.

Weaknesses:

1. The paper selects tasks from three different domains, but it is unclear whether the chosen methods are mainstream approaches within their respective fields. This raises concerns about whether the proposed method has been fairly compared with widely recognized and robust baselines. For example, in molecular dynamics (MD) simulations, the paper employs a temporal equivariant graph network to predict atomic positions in future frames. While this is a valid mathematical modeling choice, a more common approach in MD simulations is to first predict atomic forces at each frame and then compute the next frame’s positions accordingly to better preserve physical consistency. If the paper does not intend to directly predict forces like machine learning force fields in MD, it should provide an explanation for this choice.

2. The choice of baselines may not be comprehensive enough, particularly for domain-specific models (typically listed above the first horizontal divider in tables). For instance, in Table 1, EGNN represents work from 2021—should more recent methods from the past two years, such as Equiformer v2, MACE, etc., be included to strengthen the credibility of the results? Similar concerns apply to other tasks as well.

3. The integration of a large language model inevitably leads to a significant increase in inference time. However, the results section does not provide any data regarding model parameters, training time, or inference time, making it difficult to assess the computational cost of this combination. Given this, in the selection of baselines, it is reasonable for the proposed method to outperform large language models alone (typically listed between the first and second horizontal dividers in tables) since additional information and parameters are introduced. However, in domain-specific tasks such as molecular dynamics prediction and protein structure prediction, there are already many large-scale pretrained models, such as Uni-Mol, AlphaFold, etc. If inference time is not considered a primary factor, should these methods also be included in the comparison?

[1]Liao Y L, Smidt T. Equiformer: Equivariant graph attention transformer for 3d atomistic graphs[J]. arXiv preprint arXiv:2206.11990, 2022.

[2]Liao Y L, Wood B, Das A, et al. Equiformerv2: Improved equivariant transformer for scaling to higher-degree representations[J]. arXiv preprint arXiv:2306.12059, 2023.

[3]Batatia I, Kovacs D P, Simm G, et al. MACE: Higher order equivariant message passing neural networks for fast and accurate force fields[J]. Advances in neural information processing systems, 2022, 35: 11423-11436.

[4]Ji X, Wang Z, Gao Z, et al. Uni-Mol2: Exploring Molecular Pretraining Model at Scale[J]. arXiv preprint arXiv:2406.14969, 2024.

[5]Jumper J, Evans R, Pritzel A, et al. Highly accurate protein structure prediction with AlphaFold[J]. nature, 2021, 596(7873): 583-589.

**Questions For Authors:**

1. In the Equivariant Adapter section, the paper describes \( e_{\phi_m}, \phi_x, \) and \( \phi_h \) in Equation (6) as Multi-Layer Perceptrons (MLPs), but their specific dimensions do not appear to be provided in the main text or appendix. Have the authors considered adding these details in the appendix or releasing the source code to enhance reproducibility?

2. In the ablation study, the paper analyzes the impact of including the Equivariant Encoder, but since the LLM itself is not equivariant, only the invariant part of the output can be fed into the LLM. Can we demonstrate that the invariant features obtained through the Equivariant Encoder truly capture "spatial information" or similar properties that improve LLM predictions, compared to a potential Invariant Encoder? Given that invariant computations are generally more computationally efficient than equivariant ones.

**Relation To Broader Scientific Literature:**

No.

**Theoretical Claims:**

See subsequent subsections.

---

> ### Author Rebuttal · Authors · 2025-04-01
>
> We sincerely thank you for the time and careful consideration you have given to providing detailed and constructive feedback. Your valuable insights have greatly improved both the technical accuracy and clarity of our manuscript. We have meticulously revised the paper to incorporate your suggestions. Below, we respond to each of your comments point by point.
>
> ## Responses to Weaknesses
>
> > **Q1: The explanation for this choice.**
>
> In our experiments, we follow the mainstream settings on three different domains. For MD17 and Human Motion Capture datasets, future frame prediction is a common setting that used in many studies, such as Eqmotion[1] and ESTAG. This approach offers a more direct method to assess model capabilities without relying on external solvers. For the antibody design task, we benchmarked against MEAN (ICLR 2023) and GeoAB (ICML 2024), which represent the current mainstream and state-of-the-art approaches in this domain.
>
> > **Q2： Including additional baselines.**
>
> Thank you for your suggestion. Due to the tight time limitation, we have included additional experimental results for only one recent model, Equiformer, on the MD17 dataset. The results in Table F indicate that our EquiLLM still outperforms Equiformer on by a large margin. We'll include more experiments across three tasks to further strengthen credibility.
>
> **Table F. The performance of Equiformer on MD17.**
>
> ||Aspirin|Benzene|Ethanol|Malonaldehyde|Naphthalene|Salicylic|Toluene|Uracil|
> |-|-|-|-|-|-|-|-|-|
> |Equiformer|10.13|2.00|1.88|8.05|3.43|5.79|2.09|4.38|
> |EquiLLM|**2.391**|**0.732**|**1.031**|**1.671**|**1.453**|**2.162**|**1.178**|**1.060**|
>
> > **Q3: Computational cost & Include the Uni-Mol, AlphaFold, etc.**
>
> We appreciate your comments. As shown in Table G on antibody desgin task, our comparative analysis of inference times reveals that EquiLLM requires slightly more computation than state-of-the-art  methods (MEAN and GeoAB), but this modest overhead is justified by its substantial accuracy gains. However, it is worth to note that the primary contribution of this paper is not the optimization of computational cost, but rather the integration of Pretained Large Language Models into geometric learning. We excluded large pretrained models (Uni-Mol, AlphaFold) due to their extensive domain-specific pretraining and significantly higher computational costs, which would result in an unfair comparison.
>
> **Table G. The inference time on RAbD.**
>
> ||Inference time/s|
> |-|-|
> |GeoAB|0.0265|
> |MEAN|0.0139|
> |EquiLLM|0.0539|
>
> ## Responses to Questions For Authors:
>
> > **Q1: More details**
>
> Great suggestion! We will include detailed descriptions of the MLP dimensions in the manuscript's appendix and will open-source the code upon paper acceptance.
>
> > **Q2: Do Invariant features capture "spatial information"**
>
> Thank you for your question. In our Equivariant Encoder, equivariant and invariant features interact through message passing and feature updating, with 3D spatial distances explicitly encoded. As established in PAINN (Section 3.3), incorporating distance information across stacked layers implicitly models angular relationships, enabling the output invariant features to inherently capture spatial geometric information.
>
> To validate our design, we have conducted ablation studies by replacing the Equivariant Encoder with two types of invariant encoders: (1) a standard GNN (see Table A, Response to Reviewer c49n) and (2) a canonicalization approach converting equivariant vectors to invariant forms (see Tables D&E, Response to Reviewer M747). Both variants underperformed our original model, confirming that the advantage of equivariant encoder against the invariant counterparts in effectively capturing spatial information.
>
> [1] EqMotion: Equivariant Multi-agent Motion Prediction  with Invariant Interaction Reasoning, CVPR2023.

---

### Official Review · Reviewer_M747 · 2025-03-13

**Overall Recommendation:** 3

**Summary:**

This paper presents a method for solving equivariant tasks by combining a pre-trained large language model (LLM) with a trained, geometric graph network. The large language model is prompted only with invariant quantities, which come from both a natural language prompt and learned invariant features from the graph network. It outputs invariant quantities, which are fed back into a new equivariant graph network. Only the equivariant networks are trained, while the LLM weights are frozen. They evaluate their method on a molecular dynamics dataset, a human motion capture dataset, and an antibody design dataset.

**Claims And Evidence:**

The claims made regarding experimental performance, relative to the chosen baselines (more on that later), are clearly supported by the reported numbers. However, I found several of the motivating claims to be made without evidence/citation. For example:

“A natural idea is to directly employ LLMs for modeling 3D physical systems. However, this approach fails to yield
satisfactory results in practice.” Are there citations to support this?

“A key limitation is that LLMs are trained to process ordered and discrete text tokens, restricting their ability to directly comprehend unordered and continuous data in 3D space.” Actually, tokenizing 3D structures is an active area of research, and has been deployed successfully in recent papers e.g. ESM3, ProSST, BindGPT, CHEAP, Geo2Seq, etc.

“Therefore, it is non-trivial to integrate the strengths of both LLMs and geometric GNNs while maintaining essential geometric properties.” Canonicalization is a very natural way of achieving this, which has been used together with LLMs for certain applications. Thus, I find this claim too strong. This should also be added as a baseline.

“More significantly, LLMs’ flexibility in prompt engineering enables the development of tailored instructions that better leverage their capabilities, producing outputs more precisely suited to the task.” I believe that this is probably true, but for good scholarship, statements like this should either be phrased as “We speculate, based on our results, that…” or with explicit citations to back up the claim.

"Although the aforementioned methods promote interactions between GNNs and LLMs through various paradigms and yield promising results, they have yet to explore tasks involving 3D structural data, such as 3D structure generation and dynamic trajectory simulation in 3D space.” I believe that this is simply not true; consider e.g. ESM3. Although it uses equivariant attention instead of an equivariant graph network, I think this is tangential to the claim. The authors should tone down the strength of this claim, e.g. perhaps if you restrict to works which freeze pre-trained LLMs this is true (although, with the sheer volume of LLM literature, it is hard to say for absolutely sure).

**Essential References Not Discussed:**

There are several papers that combine language models with equivariant layers that are not discussed. For example, the ESM3 paper trains (from scratch) a masked language model that includes equivariant modules for the 3D structure channel, as well as other channels containing non-structural information (similar to this paper’s task prompt).

**Experimental Designs Or Analyses:**

It seems to me that the lack of other, stronger baselines are the biggest flaw in the experimental design. For example, natural ones include fine-tuning the LLM (without a geometric module), canonicalizing the inputs to the LLM, etc. The authors claim that things like fine-tuning are “too expensive”, but it is also not fair to compare their method (which involves some training computation as well as a pretrained LLM) to methods which have no training compute (pretrained LLM alone without fine-tuning) or to methods which have no access to a pretrained LLM (such as geometric models trained from scratch).

As a useful sanity check, I would recommend computing the equivariance error for each model, as a way of ensuring that there are no implementation bugs in the proposed method.

One ablation I would like to see is, replacing the equivariant graph network with a non-equivariant network (eg a transformer).

**Methods And Evaluation Criteria:**

The benchmark datasets do make sense, and they cover a diverse range of tasks.

**Other Comments Or Suggestions:**

It is straightforward to use pertained, non-equivariant models for equivariant tasks with canonicalization and/or frame-averaging (see e.g. “Equivariant Adaptation of Large Pretrained Models” by Mondal et al 2023). The use of a trained geometric module is strictly more general, so it might perform better, but the authors should check this by comparing their method to canonicalization (perhaps allowing the same computation budget to fine-tune as was used to train the graph networks).

The clarity of the paper could be strongly improved along certain dimensions. For example, it is not made clear which parts of the learning pipeline are actively trained vs pretrained (and fixed) vs fine-tuned, except for a very brief aside on Line 177 that the LLM weights are frozen. This is a very important part of the proposed method and should be made clear from the start.

The related work also needs work. I believe that “Geometric GNNs” is too broad of a category to properly summarize in one paragraph; the authors cite a seemingly random selection of specific papers instead of summarizing overall categories of approaches (and then citing multiple papers for each paradigm). Related work should summarize the state of a field to the extent relevant for the contribution, which the related work currently does not do. Papers such as ESM3 and others, which use language models for structural tasks, are also not adequately cited. Also, ESTAG is one of the main comparison methods, so it should be described in greater detail in the experiments section.

Overall, I think the idea is neat and intuitive, and I'd like to see a more polished version of this paper (with more thorough baselines), published eventually.

Some typos:
*L16, “fall” -> “fail”
*L292: “definite”

**Other Strengths And Weaknesses:**

Strengths:
The proposed method is easy to understand, and it does not seem too hard to implement since only the geometric graph network is trained (not the LLM). It can be adapted to a variety of domains, as shown in the experiments. Also, the performance is quite a bit better than the chosen baselines. I think this merging of LLMs with geometric methods is a valuable direction.

Weaknesses:
The idea itself is simple, and feels like an incremental change relative to the literature — yet the paper is framed as a methods paper, not as an application paper. The baselines/ablations are not a very strong comparison: the pure graph models do not get to benefit from a pretrained language model in any way, whereas the language models are not fine-tuned at all on the task (which is not the case for e.g. Gruver et al 2024, which fine-tunes a pretrained language model for materials generation). It is of course good to compare EquiLLM to graph networks trained from scratch, but stronger baselines are necessary to validate the authors’ specific method. Also, the authors make very strong claims about the novelty of their work, which ignores existing, more complicated methods that train combination language models and geometric modules from scratch, together (eg ESM3); the related work and contextualization of this work is lacking.

**Questions For Authors:**

1. How can you ensure that there wasn’t data leakage, where the chosen datasets were used to train the LLMs (both the one that is used in your method, and the ones you compare against)?

2. Can the authors please contextualize their experimental results (specifically, the evaluation metrics) by citing the SOTA numbers for each task, with references?

**Relation To Broader Scientific Literature:**

The lines of work on equivariant architectures, vs language model approaches, are mostly disparate; this paper unifies them in a way that’s conceptually easy to understand.

**Theoretical Claims:**

n/a

---

> ### Author Rebuttal · Authors · 2025-04-01
>
> We are deeply grateful for the time and effort you have dedicated to offering valuable feedback. We have revised the paper to address all your comments. Below, we respond to each of your points in detail.
>
> **Questions in Claims And Evidence**
>
> > Q1:
>
> In our experiments, we evaluated multiple LLMs (GPT, Gemini, and DeepSeek) by directly inputting 3D systems, all of which performed significantly worse than our method. This finding aligns with CrystalLLM's [1] conclusion that direct fine-tuning of LLMs for 3D structure modeling leads to suboptimal performance across 5/8 evaluation metrics.
>
> > Q2
>
> While our work focuses on equivariant 3D tasks requiring directional vector outputs, the works you mentioned primarily address invariant 3D tasks. That said, we will expand our discussion in the revised manuscript to incorporate recent advances in 3D structure tokenization.
>
> > Q3
>
> We have expanded our discussion of canonicalization in the manuscript and moderate the original claim accordingly.
>
> > Q4
>
> We have revised the claim in our paper in the format as you suggested.
>
> > Q5
>
> Regarding the claim in the Related Work section, our intention was primarily to contrast our EquiLLM with the LLM+GNN approaches mentioned in the same paragraph. In the revised version, we will add a detailed discussion of ESM3 and removed the original claim to ensure a more precise and rigorous presentation.
>
> **Questions in other sections:**
>
> > Q6: Sanity check.
>
> We have computed the equivariance error for each module, and the overall model indeed satisfies equivariance.
>
> > Q7: Ablation  study.
>
> We have replaced the equivariant GNN with a normal GNN, with the results shown in Row 2 of Table A in our response to Reviewer c49n. The model exhibits performance degradation, indicating the importance of maintaining E(3) equivariance
>
> > Q8: Lack of other, stronger baselines.
>
> As requested, we have conducted the following experiments:
>
> 1. Using pretrained models with canonicalization: On the MD17 dataset, we first subtract the mean from coordinates to ensure translational invariance, then perform SVD decomposition for rotational invariance. We directly feed this canonicalized data into GPT-4o-mini, with results shown in Row 2 of Table D. The results demonstrate that while canonicalization indeed improves model's predictive capability, there remains a remarkable performance gap compared to our EquiLLM. This suggests that direct prediction of 3D coordinates remains suboptimal for current LLMs.
>
> Table D. Results of the pretrained models with canonicalization on MD17.
>
> ||Aspirin|Benzene|Ethanol|Malonaldehyde|Naphthalene|Salicylic|Toluene|Uracil|
> |-|-|-|-|-|-|-|-|-|
> |GPT-4o-mini|13.070|9.581|5.011|9.910|35.155|10.627|8.132|9.762|
> |GPT-4o-mini+canonicalization|11.783|3.055|4.512|8.916|8.263|9.751|6.364|8.989|
> |EquiLLM|**2.391**|**0.732**|**1.031**|**1.671**|**1.453**|**2.162**|**1.178**|**1.060**|
>
>
> 1. Fine-tuning the LLM: Following CrystalLLM, we finetune the llama-7b model on MD17. Due to token length limitations in our prediction task (predicting 10 frames), we select the smallest molecule, Ethanol (3 heavy atoms) for evaluation. We have investigated three experimental settings: (1) 500 samples (the original paper's setup) trained for 10 epochs; (2) 30,000 samples trained for 1 epoch; (3) 30,000 samples trained for 1 epoch with canonicalization. The results in Table E reveal that without canonicalization, 500-sample and 30,000-sample fine-tuned models perform poorly, lagging behind EquiLLM by two orders of magnitude. Remarkably, when we incorporate canonicalization as suggested, the model's predictive performance improved by a factor of 100, even surpassing GPT-4o-mini. This compelling result demonstrates that the combination of canonicalization with direct LLM fine-tuning is indeed promising and warrants further investigation.
>
> Table E. Results of fine-tuning LLM on MD17.
>
> ||Ethanol|
> |-|-|
> |Setting 1|460|
> |Setting 2|457|
> |Setting 3|4.446|
> |EquiLLM|**1.031**|
>
> > Q9: The clarity of the paper.
>
> In our current implementation, the LLM module remains pretrained (and fixed), while all other module parameters are learnable and trained from scratch.
>
> > Q10: The related work also needs work.
>
> In the revised manuscript, we will modify the Related Work section by providing a clearer organization, including a comprehensive discussion of ESM3 and other language model-based methods for structural tasks. We will also provide detailed descriptions of ESTAG in the experimental section.
>
> > Q11: Typos & Contextualize their experimental results.
>
> We have implemented all revisions in accordance with your suggestions in the revised manuscript.
>
> > Q12: Ensure wasn’t data leakage
>
> Our study utilizes exclusively 3D structural data, whereas all comparative LLMs were pretrained on textual corpora alone. This fundamental modality difference potentially effectively eliminates the risk of data leakage.
>
> [1] Fine-Tuned Language Models Generate Stable Inorganic Materials as Text, ICLR24

---

### Official Review · Reviewer_VBCb · 2025-03-19

**Overall Recommendation:** 4

**Summary:**

This paper puts forward EquiLLM, a strategy to merge large language models (LLMs) with geometric (E(3)-equivariant) graph neural networks (GNNs). The motivation is straightforward: GNNs with built-in physical symmetry can handle 3D data in a rotation-, reflection-, and translation-consistent way, but they typically lack the broader domain insights or contextual knowledge that LLMs are good at capturing. Conversely, LLMs excel at analyzing text and general knowledge, yet they struggle when asked to directly process 3D coordinates or enforce geometric symmetries.

EquiLLM bridges that gap by clearly splitting the workloads:
1. Equivariant Encoder (GNN) – Handles spatial structure, ensuring that rotating or shifting inputs leads to the correct rotation or shift in outputs.
2. Prompted LLM – Receives only “invariant” features and carefully prepared textual prompts (e.g., sequence data, summary statistics, or domain context). This way, the LLM can apply its pretrained knowledge without worrying about coordinate transformations.
3. Equivariant Adapter – Recombines the LLM’s outputs with the GNN’s spatial embeddings. Because the adapter is itself an equivariant module, any coordinate transformations flow through properly.

By cleanly separating invariant and equivariant representations, EquiLLM is able to inject the LLM’s knowledge about, for instance, chemical or biological concepts, while still guaranteeing correctness in handling 3D geometry. Experiments on molecular dynamics, human motion, and antibody design suggest that this setup can outperform using purely geometric networks or purely language-based models.

**Claims And Evidence:**

Overall, the key quantitative claims—namely that EquiLLM maintains E(3)-equivariance and achieves higher accuracy than both GNN-only and LLM-only baselines—are reasonably backed by the results on molecular dynamics, human motion, and antibody design. The authors provide side-by-side performance tables, ablation studies, and comparisons across multiple datasets. These empirical tests support the conclusion that adding a language model to a geometric GNN can improve 3D prediction accuracy under symmetry constraints.

However, there are a few areas where the evidence leaves some questions:

1 Role of “External Knowledge.”
   The paper attributes improvements partly to leveraging LLM “domain knowledge,” but how that knowledge is used is demonstrated only indirectly. While ablations show that prompts help, they do not isolate whether the gains come specifically from knowledge embedded in LLM pretraining or simply from a new trainable pathway (even though the LLM is frozen). A direct test—e.g., tasks that require specialized domain facts only learned from large-scale text—could clarify how much of the improvement is truly “knowledge-driven” rather than architectural flexibility.

2 Independence of Added Model Capacity.
   While the LLM’s parameters are frozen, the approach still involves an additional module (the LLM plus the adapter) beyond the geometric encoder. This extra capacity might explain some of the improvement. The authors conduct ablations highlighting the importance of prompting, but comparisons controlling for parameter count could strengthen the argument about how much improvement comes from bridging textual knowledge and 3D GNNs.

3 Formal Proofs of Equivariance.
   The paper relies largely on references to prior geometric GNN proofs and the stated separation of invariant vs. equivariant pathways. While that is common in similar research, readers not well-versed in geometric GNNs might want a more explicit derivation.

On the whole, the main findings about quantitative improvements under 3D symmetry constraints are well supported by experiments. Claims regarding “injecting domain knowledge” and attributing gains mainly to that knowledge are plausible but would benefit from more targeted evidence isolating the effect of LLM pretraining.

**Essential References Not Discussed:**

Some recently published approaches combine large language models with graph-based molecular or protein modeling but do not necessarily enforce strict 3D symmetries. Methods such as MoleculeSTM, MolCA, or Prot2Text show how text-based knowledge can be integrated with structural data. Also, newer “text-to-structure” techniques generate or modify 3D configurations directly from prompts, which might offer additional context. Finally, broader libraries such as e3nn for building E(3)-equivariant networks could situate EquiLLM among related tools for geometric learning. A brief discussion of these works would help readers place EquiLLM in the wider landscape of combining language models with 3D-aware architectures.

**Experimental Designs Or Analyses:**

The experimental setup for each of the three application domains—molecular dynamics, human motion prediction, and antibody design—largely follows established practices (e.g., using MD17 for small molecule simulations, a motion capture dataset, and standard antibody-design benchmarks). The baseline models are fairly chosen, and the evaluation metrics (root-mean-squared error for 3D positions, cross-entropy for sequences, etc.) are standard. The inclusion of ablation studies also helps show the effect of prompting and how the LLM interacts with the GNN.

One minor point is that the authors rely on a fixed set of hyperparameters across different architectures; while this is aimed at fairness, some baselines might not be fully optimized. Another is that although the ablation results highlight the model’s different components, they do not completely isolate the effect of LLM “knowledge” versus just having additional trainable modules. However, none of these issues seem to undermine the core claims, and overall the experiments appear consistent and well controlled.

**Methods And Evaluation Criteria:**

Yes, the paper’s choices of tasks and datasets are sensible for testing a framework that combines LLMs with 3D-equivariant GNNs. The molecular dynamics, human motion, and antibody design settings all demand careful handling of 3D structures under symmetry transformations, which aligns with the paper’s claim about preserving E(3)-equivariance. Plus, each of these tasks benefits from the richer contextual or domain-level reasoning that an LLM can contribute.

The benchmarks—MD17 for small-molecule dynamics, a motion-capture dataset for human skeletal movement, and a standard antibody-design dataset—are representative of real-world scenarios where both spatial invariances and contextual knowledge are key.

**Other Comments Or Suggestions:**

n/a

**Other Strengths And Weaknesses:**

Strengths:
• The paper’s main contribution—combining an equivariant GNN with an LLM by carefully separating invariant and directional information—feels fresh, especially given the clear synergy with 3D tasks.
• The authors demonstrate the technique on several real-world applications (molecular dynamics, human motion, and antibody design), indicating practical significance.
• The writing, while sometimes concise on certain technical points, is reasonably clear for readers with a background in geometric deep learning, and the experimental setup is straightforward to follow.

Weaknesses:
• The manuscript relies heavily on referencing established proofs for equivariant GNNs. A more direct or step-by-step argument that the combined system remains equivariant would improve clarity for a broader audience.
• While the experimental results are comprehensive, the paper could devote more attention to precisely how LLM knowledge influences outcomes—especially to separate general architectural effects from true “knowledge injection.”
• The ablations show the benefits of prompting but do not fully dissect which aspects of prompt engineering bring the greatest advantages.

Overall, the idea of letting GNNs handle 3D geometry and LLMs handle higher-level contextual information is an interesting hybrid design that could be valuable in a range of domains involving complex spatial structures and domain knowledge.

**Questions For Authors:**

1. How does pretrained knowledge from the LLM specifically boost performance?
2. How is E(3)-equivariance assured once data flows through the LLM?
3. Can you illustrate how “geometry-aware” prompts are formulated to remain invariant?
4. How sensitive is the method to changes in prompts or hyperparameters, especially for the LLM component?

**Relation To Broader Scientific Literature:**

This paper takes ideas from two active research areas—3D-equivariant graph neural networks (GNNs) and large language models (LLMs)—and combines them in a single method called EquiLLM. Existing geometric GNNs preserve important symmetry constraints for molecules or other 3D structures but lack broad domain understanding. Meanwhile, LLMs are trained on large amounts of text-based knowledge but do not naturally handle 3D transformations. EquiLLM addresses this by giving the LLM only invariant or “directionless” information (such as molecular descriptors or statistical summaries) while the GNN processes raw 3D coordinates. The two parts share information through a small adapter that maintains the desired geometric symmetries. Experimental results on molecular dynamics, human motion, and antibody design show that EquiLLM outperforms both purely geometric models and purely language-based models, offering a middle ground that applies each technique where it works best.

**Theoretical Claims:**

The paper primarily cites existing formulations of geometric GNNs that have already established E(3)-equivariance, rather than presenting a fully standalone proof for its combined EquiLLM framework. The core argument is that by strictly separating invariant features (handled by the LLM) from equivariant features (handled by the GNN), the architecture inherits the GNN’s established symmetry properties. Since the paper does not include a detailed, from-scratch proof of how these components integrate to preserve equivariance, there is no step-by-step proof to check in the manuscript itself. Instead, the authors rely on references to standard results in geometric deep learning.

Conceptually, the design appears consistent with known proofs for equivariant GNNs, and nothing in the method obviously breaks those symmetries. However, for a rigorous guarantee, readers would need to rely on both the cited prior proofs and a clear statement about how the LLM’s outputs (restricted to invariant inputs and outputs) blend with the GNN pipeline.

I would suggest the authors do a better job at convincing the reader that their work is theoretically sound and more self-contained.

---

> ### Author Rebuttal · Authors · 2025-04-01
>
> We sincerely appreciate your recognition of our work! We are deeply grateful for the time and thoughtful effort you have dedicated to offering such detailed and constructive feedback. Your valuable suggestions have significantly enhanced both the scholarly rigor and presentation of our manuscript. Having carefully revised the paper to reflect your comments, we now address each point in detail below.
>
> > **Q1: Role of  "External Knowledge "& how LLM knowledge influences outcomes1 Role of "External Knowledge.**
>
> Thank you for raising this point. Directly explaining how the LLM's knowledge affects the results is a challenging task. Here, we indirectly demonstrate, through ablation experiments, that the model's performance suffers significantly without properly designed prompts to activate the LLM's knowledge. Specifically, when removing antigen, light chain, and heavy chain feature descriptions from antibody design prompts (Table C, Row 1), we observe clear performance degradation, highlighting how domain-specific knowledge enhances EquiLLM's geometric modeling capabilities.
>
> > **Q2: Independence of Added Model Capacity.**
>
> We sincerely appreciate your thoughtful review. We have removed the LLM module for further ablation study as you suggested, with results presented in Row 2 of Table C. The model exhibits significant performance degradation, underscoring the critical role of LLM in our framework. We have included these ablation results in the revised manuscript.
>
> **Table C. Further ablation studies on RAbD.**
>
> ||AAR|TM-score|RMSD|
> |-|-|-|-|
> |w/o object feature|38.32%|0.9826|1.76|
> |w/o LLM|37.58%|0.9818|1.79|
> |w/o prompt1|37.84%|0.9820|1.76|
> |w/o prompt2|38.57%|0.9823|1.77|
> |w/o prompt3|38.52%|0.9827|1.74|
> |EquiLLM|**38.97%**|**0.9830**|**1.73**|
>
> > **Q3: Formal Proofs of Equivariance & How is E(3)-equivariance assured once data flows through the LLM?**
>
> We apologize for any lack of clarity in the current manuscript. To clarify, since the LLM exclusively processes invariant features, its outputs remain strictly invariant. These invariant outputs are then concatenated with the original equivariant features from the encoder through a skip connection, and subsequently processed by the equivariant adaptor. Throughout this data flow, we rigorously maintain E(3)-equivariance. In the revised version, we will add the following contents: (1) a rigorous mathematical proof of the framework's equivariance properties, and (2) a detailed analysis of how data flow maintains E(3)-equivariance throughout the architecture.
>
> > **Q4: Which aspects of prompt engineering bring the greatest advantages & How sensitive is the method to changes in prompts.**
>
> Thank you for this valuable comment! We have conducted more detailed prompt ablations on the RAbD dataset, to investigate the impact of different prompt components on model performance. For antibody design task, the object statistical information encompasses two hierarchical levels:
>
> 1. **Chain-level features**:
>     1. Inter-chain centroid distances (prompt 1)
>     2. Maximum residue-residue distances within each chain(prompt 2)
> 2. **Residue-level features**:
>     1. Statistics (max/min/mean) of residue-to-centroid distances per chain(prompt 3)
>
> As shown in Rows 3-5 of Table C, the results demonstrate that chain-level features contribute more significantly to performance improvement compared to residue-level features. We hypothesize that this discrepancy arises because chain-level features provide macroscopic structural information that better facilitates global 3D structure understanding and modeling. These comprehensive ablation results will be included in the revised manuscript.
>
> > **Q5: How  "geometry-aware " prompts are formulated to remain invariant.**
>
> Nice question! To guarantee the invariance of geometry-aware prompts input to the LLM, we exclusively employ **distance-based** statistical measures, which are inherently rotation- and translation-invariant. The three prompt types mentioned above are deliberately designed as distance metrics to preserve E(3)-invariance.

---

### Official Review · Reviewer_c49n · 2025-03-26

**Overall Recommendation:** 3

**Summary:**

The authors propose EquiLLM – a framework designed to enhance spatial reasoning in 3D structure and dynamics by integrating geometry-aware prompting and equivariant Graph Neural Network layers. Experiments on molecular dynamics, human motion, and antibody design demonstrate are carried out, and show good performance.

**Claims And Evidence:**

Most of the claims are well-supported. However, some arguments are not as convincing.
For instance, the authors “One possible solution is to adapt existing multimodal LLM architectures, such as LLaVA (Liu et al., 2024b), by treating 3D structures as a separate modality and simply replacing the image encoder with a geometric GNN. However, this naive adaptation fails to satisfy the E(3)-equivariance requirement.” It seems that what the authors do is simply replacing the encoder with an equivariant GNN encoder. I would say that the authors’ method is built on the LLaVA approach.

**Essential References Not Discussed:**

Speaking about equivariant GNN, key references such as [1] should be better discussed/acknowledged in the article.

1. Satorras, Vıctor Garcia, Emiel Hoogeboom, and Max Welling. "E (n) equivariant graph neural networks." International conference on machine learning. PMLR, 2021.

**Experimental Designs Or Analyses:**

Model comparison: it would be great to compare different encoding layers to verify the significance of the equivariance as claimed by the authors. Meanwhile, the baseline models are relatively outdated and weak models.

In line 361, “To ensure a fair comparison, all hyperparameters (e.g. learning rate, number of training epochs) are kept consistent across our model and all other baselines”. This is not a valid approach since different models would need different hyperparameters to make them function well.

**Methods And Evaluation Criteria:**

Appropriate evaluation methods are used in the article. one concern is that the authors only trained the GNN encoders and adaptors. It would be interesting to explore the impact of fine-tuning the LLM layers as well.

**Other Comments Or Suggestions:**

A comparison with LLMs trained with normal GNN encoders would be great to see the effect of equivariance here.

**Other Strengths And Weaknesses:**

Strengths: The authors have shown good empirical results, demonstrating efficacy of the proposed method.

Weaknesses: Lack of more technical details such as how is the language model trained,
The language model (GPT-2) used in the experiments is very outdated.

**Questions For Authors:**

Why is the training size so small?

**Relation To Broader Scientific Literature:**

Compared with existing Large Language Models for Sciences, incorporating equivariance GNN encoders is a meaningful attempt. And the empirical results support the motivation.

**Theoretical Claims:**

N/A

---

> ### Author Rebuttal · Authors · 2025-04-01
>
> We sincerely appreciate the time and effort you have devoted to providing detailed and constructive feedback. Your insightful comments have been invaluable in improving both the technical quality and clarity of our manuscript. We have carefully revised our paper to incorporate your suggestions. Below, we address each of your points individually.
>
> > **Q1: Method is built on the LLaVA approach.**
>
> Thank you for your comment. There may be some misunderstandings here—our method does not simply replace LLaVA's encoder with an equivariant GNN encoder, as that would compromise the framework's overall equivariance. Instead, EquiLLM introduces an innovative design, as shown in Fig.1. First, the equivariant GNN encoder extracts both equivariant and invariant features, but only the invariant features are fed into the LLM, unlike LLaVA where the LLM receives all encoder outputs. Then, after LLM processing, the output is concatenated with the encoder's equivariant features via a skip connection and passed to the equivariant adapter module to generate both equivariant and invariant predictions.
>
> > **Q2: The impact of fine-tuning the LLM layers.**
>
> Nice suggestion! We set the LLM's parameters to be trainable and fine-tune the model on the SAbDab dataset. However, the experimental results (Table A, first row) show performance degradation, suggesting that fine-tuning may compromise the original information encoded in the LLM, particularly since the dataset used for fine-tuning is not large enough.
>
> **Table A. EquiLLM with different backbones.**
>
> ||AAR|TM-score|RMSD|
> |-|-|-|-|
> |Finetune LLM|38.57%|0.9819|1.77|
> |Normal GNN encoder|32.32%|0.9308|4.14|
> |Qwen2.5-3B|**39.04%**|0.9828|1.76|
> |Original|38.97%|**0.9830**|**1.73**|
>
> > **Q3: Different encoding layers & normal GNN encoders.**
>
> Thank you for raising this point. We additionally replace the original equivariant GNN encoder with a normal GNN encoder. As shown in Row 2 of Table A, the model exhibits significant performance degradation, demonstrating the importance of maintaining E(3) equivariance when modeling 3D structures.
>
> > **Q4: Baseline models are relatively outdated and weak models.**
>
> Thank you for this constructive comment. We would like to clarify that ESTAG (NeurIPS 2023) remains the SOTA model on MD17, while MEAN (ICLR 2023) and GeoAB (ICML 2024) are also leading methods on the RAbD dataset. That said, we agree that additional baselines could further validate our approach. To address this, we have included the results of Equiformer (ICLR 2023) on MD17 in Table B, where it still significantly underperforms our model.
>
> **Table B. The performance of Equiformer on MD17.**
>
> ||Aspirin|Benzene|Ethanol|Malonaldehyde|Naphthalene|Salicylic|Toluene|Uracil|
> |-|-|-|-|-|-|-|-|-|
> |Equiformer|10.13|2.00|1.88|8.05|3.43|5.79|2.09|4.38|
> |EquiLLM|**2.391**|**0.732**|**1.031**|**1.671**|**1.453**|**2.162**|**1.178**|**1.060**|
>
> > **Q5: All hyperparameters are kept consistent.**
>
> Thank you for your comment. Our experimental settings follow the ESTAG paper, using identical parameter configurations across all baseline models to ensure fair comparisons. We also explored various hyperparameter choices for the baselines on MD17, but the performance improvements remained marginal compared to our model's results.
>
> > **Q6: Discuss/acknowledge EGNN.**
>
> Great suggestion! We will provide a more comprehensive discussion of EGNN in the revised version.
>
> > **Q7: Lack of more technical details.**
>
> Thank you for this insightful suggestion. In our EquiLLM framework, the LLM module parameters remain frozen during training. In the revised version, we will provide more technical details and a brief introduction to the specific LLM employed in our work.
>
> > **Q8: The language model is very outdated.**
>
> Thank you for your insightful observation. To address this point, we conducted additional evaluations using the Qwen2.5-3B model (see Table A, Row 3). While it shows marginal improvement in AAR, we observe slight decreases in RMSD and TM-score performance. We hypothesize that the language model's capability remains constrained by limited text-3D structure paired data; otherwise, upgrading the LLM component could yield significant gains. We leave this exploration for future work.
>
> > **Q9: The training size**
>
> Nice question! Our experimental setup primarily follows established conventions in the field. For the MD17 and Human Motion datasets, we adopt the same configurations as the EGMN and ESTAG papers, while for the SAbDab dataset, we maintain the settings used in MEAN and GeoAB.

---

### Decision · Program_Chairs · 2025-05-01

**Decision:**

Accept (poster)

**Comment:**

The paper presents a novel approach combining Large Language Models with equivariant Graph Neural Networks for 3D structure prediction tasks. The core idea of separating invariant and equivariant processing is conceptually sound. After careful consideration of all reviews and the authors' thorough response to concerns, I've decided to accept this paper. The authors have adequately addressed the initial concerns about baseline comparisons through additional experiments with canonicalization and fine-tuned LLMs, and have committed to improving the related work section. While some limitations remain, the originality of the approach, strong empirical results, and the paper's contribution to bridging geometric deep learning with language models justify acceptance. The authors should incorporate their rebuttal improvements into the final manuscript.